# Multi-compartmental MOF microreactors derived from Pickering double emulsions for chemo-enzymatic cascade catalysis

Danping Tian[1], Ruipeng Hao[1], Xiaoming Zhang [1] ✉, Hu Shi [1], Yuwei Wang[1], Linfeng Liang[2], Haichao Liu [3] ✉ & Hengquan Yang [1] ✉

Bioinspired multi-compartment architectures are desired in synthetic biology and metabolic engineering, as credited by their cell-like structures and intrinsic ability of assembling catalytic species for spatiotemporal control over cascade reactions like in living systems. Herein, we describe a general Pickering double emulsion-directed interfacial synthesis method for the fabrication of multicompartmental MOF microreactors. This approach employs multiple liquid–liquid interfaces as a controllable platform for the self-completing growth of dense MOF layers, enabling the microreactor with tailor-made inner architectures and selective permeability. Importantly, simultaneous encapsulation of incompatible functionalities, including hydrophilic enzyme and hydrophobic molecular catalyst, can be realized in a single MOF microreactor for operating chemo-enzymatic cascade reactions. As exemplified by the Grubb' catalyst/CALB lipase driven olefin metathesis/ transesterification cascade reaction and glucose oxidase (GOx)/Fe-porphyrin catalyzed oxidation reaction, the multicompartmental microreactor exhibits 2.24–5.81 folds enhancement in cascade reaction efficiency in comparison to the homogeneous counterparts or physical mixture of individual analogues, due to the restrained mutual inactivation and substrate channelling effects. Our study prompts further design of multicompartment systems and the development of artificial cells capable of complex cellular transformations.

Biological cells have evolved a perfect multi-compartment structure in the form of membranous organelles, which enables the spatial segregation or confinement of different biomolecules to perform complex metabolic processes in a highly efficient, controllable, and selective manner[1–3]. Inspired by this, tremendous efforts have been directed to the design and construction of synthetic multi-compartment systems that can closely emulate both the cellular architecture and functions[4–12]. To date, various nano- or microscale scaffolds ranging from lipid vesicles[13,14], polymersomes[15–17], hydrogels[18,19], colloidosomes[20–22] to porous solid matrixes (like silica, carbon or organic polymer et al.)[23–27] have essentially been explored as multi-compartmental nano/microreactors by successive enzymatic or chemo-enzymatic cascade reactions. Despite these intriguing advances, severe obstacles are still standing in the way of moving closer to a truly applicable cell-like microsystem. On the one hand, most of the reported multi-compartment systems involve soft matter in nature, thereby suffering from the shortfalls of poor mechanical strength and chemical robustness required by practical applications. On the other

[1]School of Chemistry and Chemical Engineering, Key Laboratory of Chemical Biology and Molecular Engineering of Ministry of Education, Shanxi University, Taiyuan 030006, China. [2]Institute of Crystalline Materials, Shanxi University, Taiyuan 030006, China. [3]Beijing National Laboratory for Molecular Sciences, College of Chemistry and Molecular Engineering, Peking University, Beijing 100871, China. ✉e-mail: xmzhang4400@sxu.edu.cn; hcliu@pku.edu.cn; hqyang@sxu.edu.cn

hand, selective permeability, which ensures biomimetic properties like membrane gating, is far less achieved till now, causing inevitable difficulty in spatial confinement of active species or uncontrollable molecular transport within the compartments. More urgently, the simultaneous and spatiotemporal programming of diverse functional components into one integrated system for executing effective multistep cascades remains a considerable challenge.

Recently, metal-organic frameworks (MOFs), which represent a broad class of crystalline porous materials that are self-assembled by metal ions/clusters and organic linkers, have risen to be promising candidates for the construction of artificial microreactors[28–33]. Ascribing to their unique properties, such a platform exhibits powerful capability to better mimic the cellular systems. For example, the facile and biocompatible assembly process, together with the rigid and adjustable framework structure, make it possible to introduce two or more different catalytic species, including enzymes, metal nanoparticles or even homogeneous molecular catalysts, into one single MOF structure[34–40]. The intrinsic microporous frameworks will provide protection and interconnected microenvironment to the active species, while enabling size-selective molecular transportation, thus rendering the positive effects of MOF matrices on catalytic activity, selectivity, and stability[42–45]. So far, several self-assembled MOF structures have been constructed to organize multiple catalytic species for operating cascade reactions[34–41,46]. However, these reported microreactors are basically limited to single compartment structures, thus resulting in lack of control over the spatial organization of different catalytic actives and inapplicability for the accommodation of incompatible catalytic transformations in one vessel. Especially, the emerging concurrent chemo-enzymatic cascades, which offer complementary means to unify the advantages of synthetic catalysts with natural enzymes, has not been implemented in a single MOF microreactor due to their mutual incompatibility stemming and distinct physicochemical properties[47,48]. Moreover, the traditional adsorption or encapsulation synthesis methods often suffer from low loading efficiency, incompact confinement or mismatching of pore size. Therefore, exploring new strategies to engineer a multi-compartmental MOF architecture and to incorporate the desired chemo- and enzymatic functionalities into one integrated system in a smart and high-efficiency way is highly pursued, but still remains an elusive task.

Herein, in this study, we develop a facile and efficient Pickering double emulsion-based interfacial synthesis method to generate hierarchically multi-compartmental MOF microreactors that can readily realize the spatial confinement of incompatible chemo- and bio-catalytic species for operating effective chemo-enzymatic cascade reactions. In sharp contrast to previous reports[49–53], this approach involves the utilization of stable Pickering double emulsions as a directing template for the growth of crystalline MOF structures, which seems to be a good choice, since such a type of emulsion provides not only large liquid–liquid interfacial areas for self-completing formation of dense MOF layers, but also independent space to introduce diverse functional species[54,55]. Moreover, benefitting from the precise manipulation capability of Pickering double emulsions, this approach allows for controllable production of MOF microreactors with unprecedentedly well-tailored inner architectures,

leading to sufficient freedom for customizing the material's structure and functional properties. As a universal method, a variety of MOF types have successfully been achieved, including MOF-74, MIL-100, MIL-88A, HKUST-1, ZIF-8, and Co-MOF-74@ZIF-67. Importantly, while this strategy does offer separate control over the inner oil and water compartments, hydrophobic organometallic catalysts and hydrophilic enzymes can be directly encapsulated simultaneously in these different compartments of MOF microreactors. Two chemo-enzymatic cascade reactions, including Grubb' catalyst/CALB lipase driven olefin metathesis/transesterification reaction and glucose

oxidase/Fe-porphyrin catalyzed oxidation reaction, were exemplified to validate this concept. Encouragingly, the multi-compartmental MOF microreactors exhibit significant enhancements in catalytic efficiencies compared with the homogeneous counterparts and physical mixture of individual analogues. We believe this strategy presented here provides a step towards the design of synthetic protocells with structural complexity and functional integration for catalytic cascades that are promising in many industrial processes.

## Results
### Design and construction of the multi-compartmental MOF microreactors

The whole synthesis process of such a MOF structure is schematically shown in Fig. 1. A two-step sequential emulsification process was first adopted to generate Pickering double emulsion on a homogenizer in the presence of two types of solid emulsifier possessing distinct particle wettability, namely hydrophilic emulsifier and hydrophobic emulsifier. Those emulsifiers were obtained easily by modifying commercial silica nanoparticles (20 nm in size, Wacker Chemie) with different organosilanes [for the hydrophilic one, $0.25\,mmol\,g^{-1}$ methyltrimethoxysilane (MTS) was used and the air-water contact angle was measured to be $75°$; for the hydrophobic one, $3.0\,mmol\,g^{-1}$ dimethyldichlorosilane (DMDC) was used and the air-water contact angle was $122°$; for detailed experiments and characterizations, see Supplementary Methods and Supplementary Fig. 1a–e of the Supplementary Information]. Initially, a primary oil-in-water (O/W) single emulsion was formulated via homogenizing a mixture of inner oil phase (n-octanol) that dissolved with organic ligands and pure water phase using relatively hydrophilic silica nanoparticles as a stabilizer (5 wt% with respect to oil). The formed O/W droplets are structurally stable, spherical in appearance with a relatively small diameter of 5 μm (Supplementary Fig. 2a–d), which were further centrifugated to high internal phase emulsion (HIPE) layer to achieve denser packing. Followed by the addition of a certain amount of metal precursor, the HIPE droplets were instantly dispersed into a stream of external oil phase that contained another portion of organic ligand with the hydrophobic emulsifier by shaking, leading to the formation of oil-in-water-in-oil (O/W/O) double emulsions. As observed by a bright-field microscopy, this complex emulsion is composed of abundant discrete micro-droplets that containing a crowd of smaller droplets inside, of which the droplet size varied from 20–40 μm for the external droplets and 3–5 μm for the internal ones (Fig. 2a₁ and Supplementary Fig. 3). The morphology and size of the inner droplets are well consistent with the primary O/W emulsions, indicating that the second emulsification process did not disrupt the initial interfacial assembly of hydrophilic emulsifiers. To better clarify the multiple structures and their stability, a series of fluorescent experiments were also performed. As shown in the confocal fluorescence microscopy (CLSM) observations (Fig. 2a₂–a₄), after dying with fluorescein Nile Red and FITC-Dextran, red (wavelength: 530 nm) and green (wavelength: 488 nm) fluorescence emissions are clearly seen in the internal droplets and dispersed aqueous zone respectively, while both the fluorescence signals are presented in the merged image, implying the successful formation of O/W/O type emulsions. Preferential location of the hydrophilic and hydrophobic emulsifiers at the internal droplets interface or external interface without apparent exchange between each other was further identified by the fluorescence dyeing experiments using fluorescein isothiocyanate isomer I (FITC-I) and Rhodamine B labeled silica as emulsifiers, which showed the appearance of fluorescent circles around the droplet surfaces in the 2D or 3D CLSM images (Fig. 2b₁–b₄ and Supplementary Fig. 4). These irreversibly adsorbed silica emulsifiers would provide efficient protection towards the dispersed droplets, thus endow them with long-term structural stability. As verified by a standing test, the obtained double emulsions can be storage for several days without apparent coalescence or phase separation

(Supplementary Fig. 5), which is long enough for the following interfacial growth of crystalline MOF structures. Furthermore, to better explore the excellent stability of the Pickering double emulsions, we also used fluorescence microscopy to visualize the possible migration of confined molecules between the internal and external oil phases. As shown in Fig. 2c, when a hydrophobic probe molecule, Nile Red, was preloaded within the internal oil droplets, its red fluorescence signals always existed within the internal droplets over the whole monitored time, suggesting that the dye molecules did not diffuse through the aqueous phase to enter the external oil surrounding. If this dye molecule was initially dissolved in the external oil phase, almost no fluorescence signal was detected within the internal droplets (Supplementary Fig. 6). Similarly, as the hydrophilic FITC-Dextran was preloaded, green florescence signal was only observed within the aqueous zone (Supplementary Fig. 7). These results sufficiently demonstrate the positional compartmentalization ability of the double emulsions and limited molecular migration between different phases. Then, after the formation of Pickering double emulsion droplets, the interfacial growth of MOF layers began to take place on the basis of metal coordination interactions. In this case, the organic ligands are available in the internal oil droplets and external oil surroundings, while the metal ions are existed in the dispersed aqueous phase, herein it is expected that the crystallite formation occurs from the inside/outside directions of the multiple liquid–liquid interfaces. Since such an interfacial growth process is intrinsically self-completing, crystallites will be preferentially formed at the defects remaining in the interfaces, so tend to the formation of dense and defect-free MOF layers on the interfaces[49,50]. Moreover, the free-standing MOF skeletons adopted the shape and complex structure of the multiple interfaces during synthesis, thus this strategy not only provides an access to converting dynamic emulsion systems into robust MOF structures, but also enables the resulting skeleton to be shaped as multi-compartmental objects.

To verify this methodology, we initially focused on the synthesis of a representative material, MOF-74, using $Ni(NO_3)_2 \cdot 6H_2O$ and 2,5-dihydroxy-1,4-benzen dicarbon-xylic acid ($H_4DOBDC$) as precursors. Scanning electron microscopy (SEM) images reveal the formation of intact spherical structures with a particle size distribution of $30 \pm 10$ µm (Fig. 2d and Supplementary Fig. 8a). The external surfaces of these microspheres are relatively rough and covered by a layer of silica nanoparticles (Supplementary Fig. 8b), further confirming the interfacial assembly of solid emulsifiers. The hierarchically multi-compartmental architecture is clearly visible from an individual cracked particle (Fig. 2e), where a crowd of sub-compartments are sealed within an appressed outer crust. For the interior structure, one can see the presence of packed hollow capsules with a cavity diameter of 3–5 µm (Fig. 2f), which is consistent well with the original size of O/W droplets, indicating the template action of these droplets. A magnified cross-sectional view of the outer boundary shows that the interfacially formed MOF layer is dense and continuous with a thin thickness of 50–60 nm (Fig. 2g). To further identify the MOF layer structures, transmission electron microscopy (TEM) observation was also performed. As shown in Fig. 2h and Supplementary Fig. 8c, although the internal or external MOF layers can't be distinguished from each other, a compact and defect-free crystalline MOF structure with decorated silica nanoparticles is still clearly observed from any

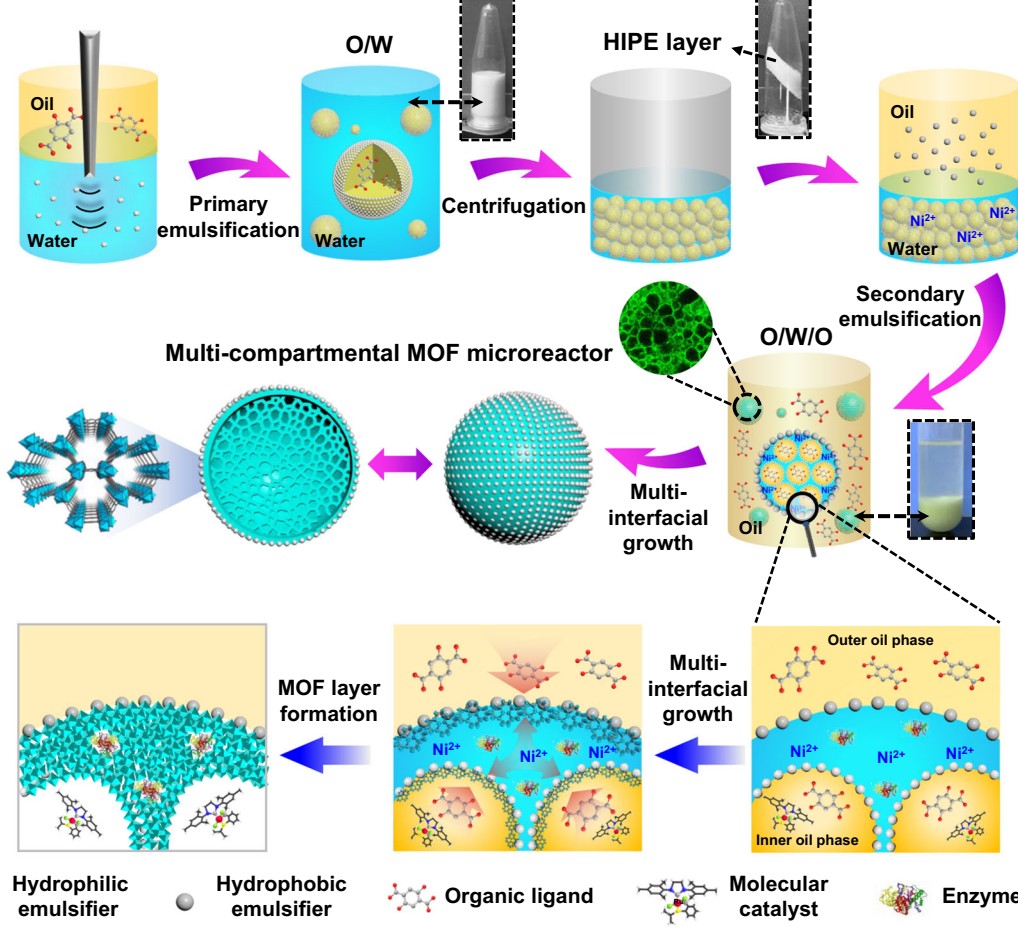

**Fig. 1 | Schematic illustration of the preparation of multi-compartmental MOF microreactors.** Pickering double emulsions were formed via a two-step emulsification process, the organic ligands dissolved in the interior and outer oil phase coordinated with metal ions at the multiple liquid–liquid interfaces.

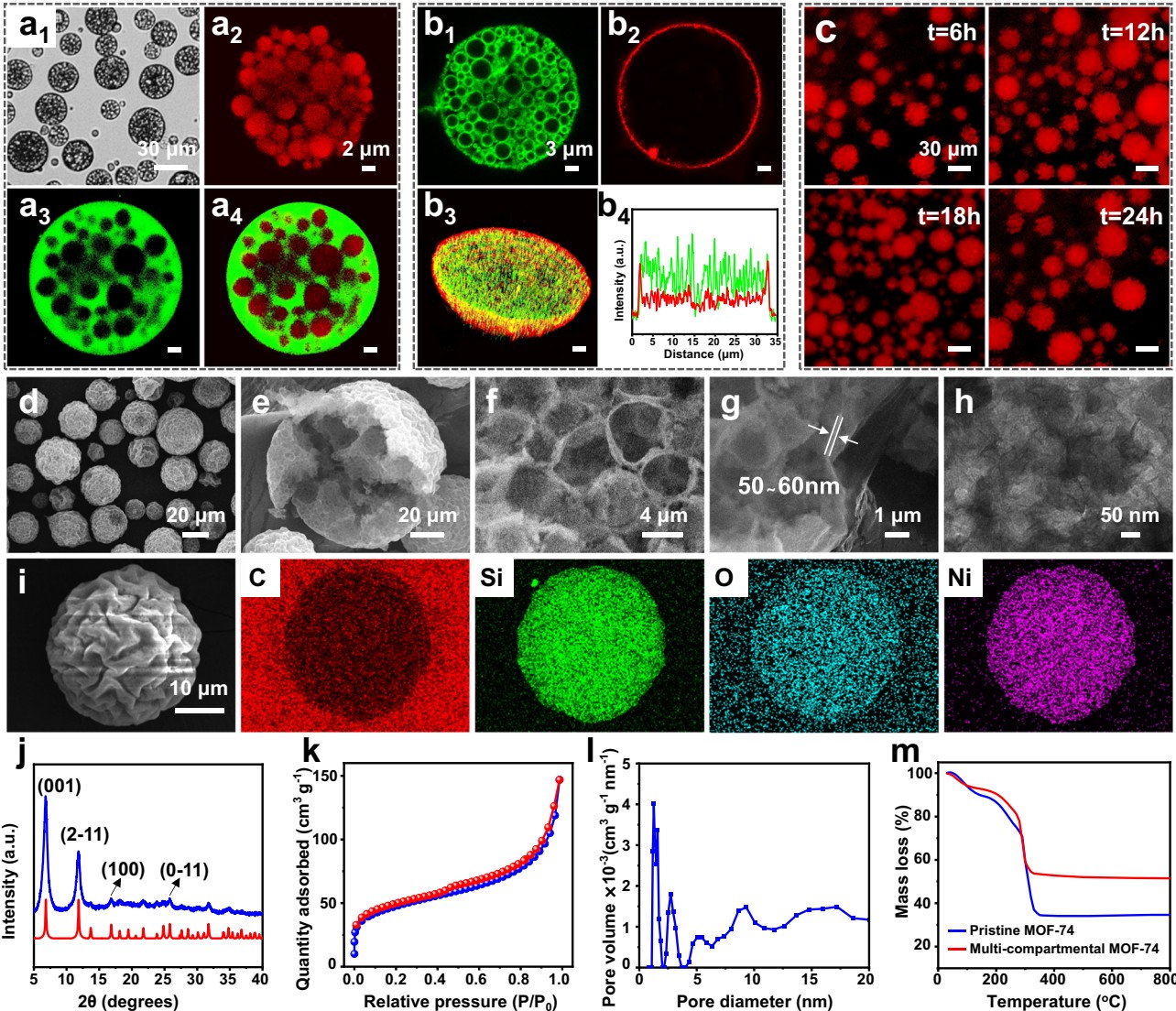

**Fig. 2 | Characterizations of the Pickering double emulsion and its derived multi-compartmental MOF-74 microreactor. a₁–a₄** Bright-field and confocal fluorescence microscopy images of the emulsion droplets with inner oil and water phases dyed by Nile Red and FITC-Dextran respectively [observed at 530 nm (**a₂**), 488 nm (**a₃**) and dual-wavelength (**a₄**)]. **b₁–b₄** Confocal fluorescence microscopy images of a single droplet with hydrophilic silica emulsifier labeled by FITC-I (**b₁**), hydrophobic silica emulsifier labeled by Rhodamine B (**b₂**), a 3D merged image (**b₃**) and its corresponding line profile of fluorescence intensity across the droplet (**b₄**).

**c** Fluorescence observation of the droplets dyed by Nile Red as a function of storage time. **d–g** SEM images of the multi-compartmental MOF-74 microreactors, a low magnification observation (**d**), a single damaged particle (**e**), a magnified observation of interior multi-compartment structure (**f**) and a magnified boundary observation (**g**). **h** TEM image of the formed MOF-layer. **i** A single particle and its elemental mappings of C, Si, O, and Ni. **j** XRD pattern. **k, l** N₂ sorption isotherms and pore size distribution. **m** TGA curve.

casual observations. Moreover, we found that the dosage of coordination precursors directly determined the formed MOF skeletons at the interface, leading to distinct inner and outer MOF layers with varied thicknesses (Supplementary Fig. 9). Elemental mappings indicate the homogeneous distributions of C, Si, O, and Ni in the whole microparticles (Fig. 2i). The Fourier transform infrared spectrum (FT-IR) shows the characteristic stretching frequency of carboxylate with a red shift compared to uncoordinated H₄DOBDC, confirming the coordination of metal ions and organic ligand (Supplementary Fig. 10). X-ray powder diffraction (XRD) of such a material shows obvious characteristic peaks at 8.2° (001), 12.2° (2–10), 16.9° (100), and 25.8° (0–11), which is in good agreement with that of the simulated pattern of MOF-74 (Fig. 2j), suggesting the successful formation of aimed crystalline MOF structure. Nitrogen sorption analysis was also performed to validate the porosity of our MOF-74 superstructure. As shown in Fig. 2k, a nonclassical type-IV isotherm curve featuring with

an apparent increasing N₂ uptake at low relative pressure of 0–0.03 and an unsaturated adsorption capacity at high relative pressure of >0.95 is exhibited, indicating the coexistence of hierarchical pores in this material. The pore size distribution calculated from DFT method further confirms the coexisted micropores (-1.3 nm) and mesopores, that are arising from the distinct pores of MOF layer and interstices between particle packing (Fig. 2l). As for the unlevel adsorption quantity, it can be ascribed to the macropores from interior hollow cavities. The Brunauer-Emmett-Teller (BET) surface area and pore volume of this material are determined to be 166.6 m² g⁻¹ and 0.22 cm³ g⁻¹ respectively. These textural properties are in the lower range of values reported for pristine MOF-74[56,57], which is most likely related to the inorganic silica decorated hybrid structure. Furthermore, to reveal the component difference and thermal stability of our MOF-74 super-structure compared to its pristine counterpart, thermogravimetric analysis (TGA) was conducted. As exhibited in Fig. 2m,

although these MOF-74 materials are thermally stable up to 300 °C and began to decompose in a single step upon further increasing the temperature, their residual mass loss is distinctly different (54.6% vs 34.5%), due to the presence of silica emulsifiers within the multi-compartmental MOF-74 superstructure.

## Manipulation of the interior architectures

In fact, apart from the described structural stability, our Pickering double emulsion also features controllable ability over its inner compartments, which enables us to tailor the resultant MOF materials with finely manipulated structural evolutions. To achieve this, we independently tuned the inner droplet size by changing the dosage of hydrophilic emulsifier as well as regulated the inner droplet occupancy by adjusting its volume fraction. Optical micrographs of the primary O/W and final O/W/O emulsions show that the average diameter of inner droplets gradually increased from 1.9 to 7.8 and 10.2 μm, while reducing the dosage of hydrophilic emulsifier from 10 to 2.5 and 0.8 wt% with respect to oil phase (Supplementary Fig. 11). This variation allows us to engineer the size of interior capsular structures predictably. As shown in Fig. 3a–c, similar multi-compartmental microspheres with different sized sub-compartments ranging from 1.7 to 7.5 and 10.0 μm are obtained from those emulsions. Notably, in these cases, the volume fraction of O/W emulsion (with respect to the dispersed water phase) is remained constant as high as 75%, therein their interiors are all whole filling with sub-compartments. Interestingly, we found that the utilized volume fraction of inner droplets not only determines the occupancy of sub-compartments, but also has a more dramatical impact on the interior architecture of the resultant MOF material. By increasing the fraction from 75% to 80%, a denser honeycomb-like structure with close packing interconnected compartments is achieved (Fig. 3d, Supplementary Fig. 12a), and the mean diameter of the inner sub-compartments remains unchanged. However, when decreasing the fraction to 70%, the occupancy of inner compartments becomes less, while the interior gradually turns hollow from the area close to the center to that adjacent to the microwells (Fig. 3e, Supplementary Fig. 12b). This phenomenon was more pronounced for the partially occupied hollow structure by further decreasing the volume fraction to 65% (Fig. 3f, Supplementary Fig. 12c). Moreover, it was found that discrete well-defined capsules will be formed, if an additional pre-coordination process of the inner droplets was performed before the secondary emulsification, as demonstrated by the abundant encapsulated hollow microparticles in SEM images (Fig. 3g, Supplementary Fig. 13a). In this situation, however, upon decreasing the volume fraction of primary O/W emulsion, the extent of inner separate capsules decreased successively, and the overall spherical morphology began slumping, as reflected by the less capsule numbers and crumpled structures (Fig. 3h, i and Supplementary Fig. 13b, c).

Given such a controlled manipulation of the interior architectures, we are aware of the underlying mechanism for the structural evolution (Fig. 3j). Generally, there are three different interactions that dominate the final interior structure of the MOF materials: (i) van der Waals attraction between droplet surfaces, (ii) intrinsic migration interaction of inner droplets arising from the Laplace pressure associated with the interfacial tension and (iii) coordination interactions between metal and organic ligand precursors[58–60]. At high volume fraction of O/W droplets (>75%), the gel-like HIPE within the aqueous droplets is highly viscous and tend to aggregate closely due to the driving forces arising from i and ii, as verified by the visible observations using a laser confocal microscope (Fig. 3a$_1$–c$_1$). Moreover, since the fast coordination process of metal ions and organic ligands (Supplementary Fig. 14), a solid MOF layer is formed quickly at the inner O/W and outer W/O interfaces (interaction iii), which further rigidifies the thermodynamically unstable droplets. Following the continuous interfacial growth of crystals, a robust MOF skeleton with interconnected tissue-like structure can be produced. In this specific case, one can predict that the space size of inner compartments is flexible by altering the inner droplet size. By contrast, while decreasing the volume fraction of O/W droplets to 65%, the content of inner droplets is far from complete filling, and the existed interactions i and ii lead the O/W droplets aggregate around the microwells, forming toroidal microstructures. This is further demonstrated by the moderate fraction of 70%, in which a broader microwells are occupied from the walls to the center. Notably, the assembly of inner droplets seems critical for structuring the hierarchical MOF entity to produce intact microspherical particles. If no primary emulsion is applied, a single crumpled hollow structure will be formed, since the absence of supportive inner frameworks (Supplementary Fig. 15). Interestingly, when a pre-coordination process of the inner droplets is performed, the adhesive interaction between different droplets can be weakened due to the altered surface properties. Therein, the interconnected internal structures gradually become mutually isolated and retain the integrity of their compartmentalized interior, along with the increase of pre-coordination time from 1 to 4 h (Supplementary Fig. 16). In this case, less inner capsules without apparent adhesion are produced after reducing the volume fraction of O/W droplets. To get more insight into the formed MOF layer, TEM observations of the inner compartments are also performed. As shown in Supplementary Fig. 17, discrete hollow microspheres with a dense and uniform crystalline MOF shell (thickness is ~100 nm) are clearly formed, further confirming the superior advantages of our multiple interfaces directed self-completing synthesis method.

## Versatility and Scalability

To explore the versatility of our strategy, we extended this approach to the synthesis of a series of structurally diverse MOFs, including Ni-MOF-74-II (Ni(NO$_3$)$_2$·6H$_2$O and 4,4′-dihydroxybiphenyl-3,3′-dicarboxylic acid as precursors), MIL-100 (FeCl$_3$·6H$_2$O and 1, 3, 5-benzenetricarboxylic acid as precursors), MIL-88A (FeCl$_3$·6H$_2$O and fumaric as precursors), HKUST-1 (Cu(NO$_3$)$_2$·3H$_2$O and 1, 3, 5-benzenetricarboxylic acid as precursors), ZIF-8 (Zn(NO$_3$)$_2$·6H$_2$O and 2-methylimidazole as precursors) and Co-MOF-74@ZIF-67 (Co(OAc)$_2$·4H$_2$O, H$_4$DOBDC and 2-methylimidazole as precursors). Fig. 4ii–iv presents typical SEM images of the resulting MOF materials, in which individual microspheres with multi-compartmental interior structures are achieved. Specifically, the interior structure of ZIF-8 microreactor is distinct from other types of MOF materials [Fig. 4e(iii, iv)], since the solubility of its utilized 2-methylimidazole is quite different from other organic ligands, which could be dissolved in both aqueous and organic phases. The XRD patterns of these materials match well with the simulated ones (Fig. 4v), verifying the formation of targeted MOF crystal structures. Notably, this method can also be applied for the synthesis of hybrid MOF systems by using different organic linkers. Taking Co-MOF-74@ZIF-67 as an example, H$_4$DOBDC and 2-methylimidazole are respectively dissolved in the inner oil droplets and outer oil continuous phase, so tend to coordinate with the metal ions at the internal and external interfaces, forming hybrid MOF composites as confirmed by the mixed XRD pattern. For these materials, further characterization by energy dispersive X-ray (EDX) elemental mapping (Supplementary Fig. 18), FT-IR (Supplementary Fig. 19) and N$_2$ physical adsorption (Supplementary Fig. 20) confirmed the formation of the corresponding MOF skeletons with relatively good porosity, and all calculated BET surface areas (Supplementary Table 1) were lower to previously reported values due to the presence of silica emulsifiers[52,57,61–63].

Remarkably, our Pickering double emulsion templated technique can also be easily scaled up, which is critical for their implementation in real-life applications. As verified in Supplementary Fig. 21, large-scale production (emulsion volume 250 mL, product mass 14.8 g) of the multi-compartmental MOF-74 materials without any change of their morphology, interior structure and crystalline frameworks were

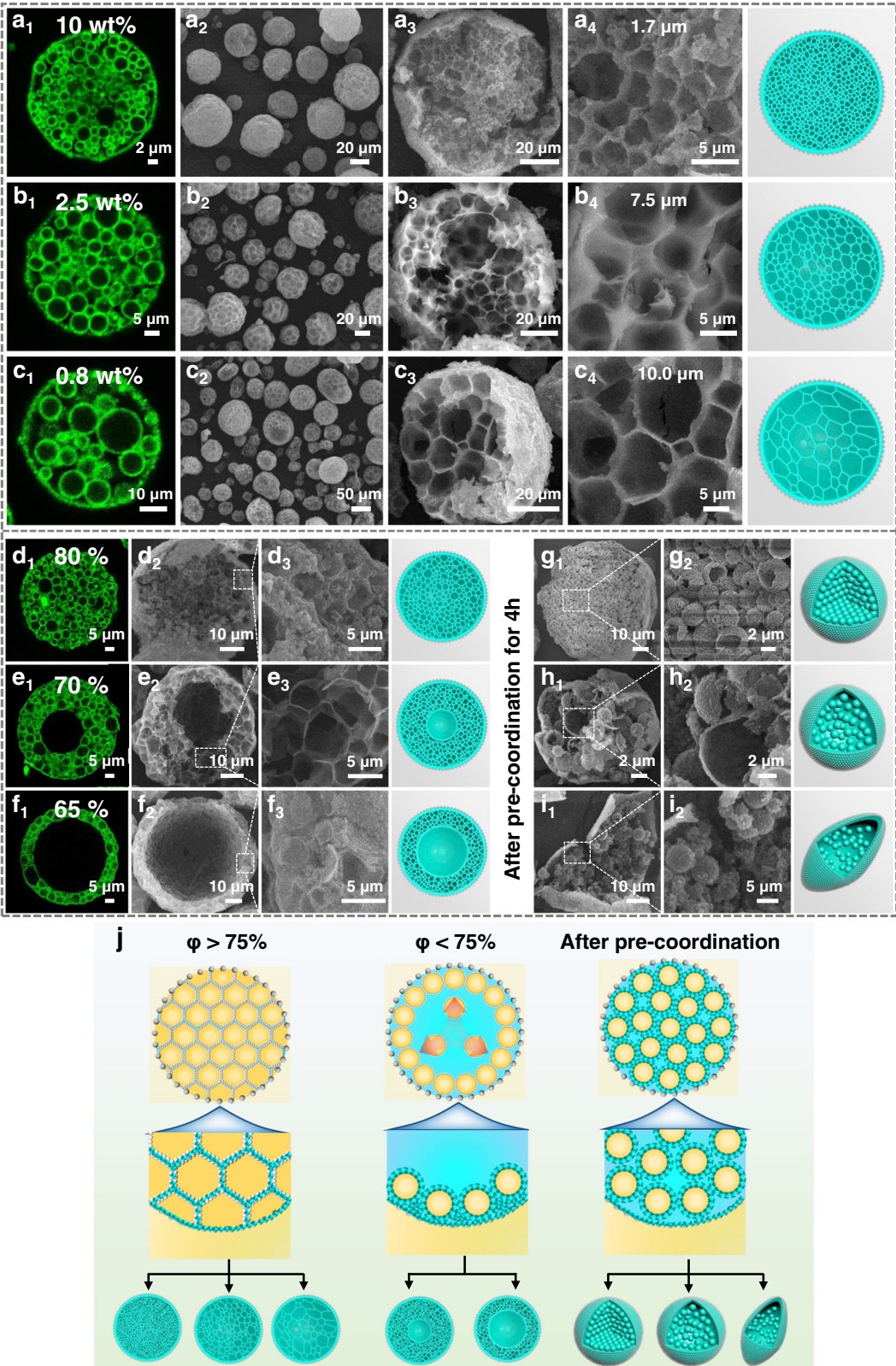

**Fig. 3 | Structural evolution of the multi-compartmental MOF-74 microreactor.** **a–c** Fluorescence images of the original emulsions with different hydrophilic emulsifier dosages and SEM images of their derived microstructures with varied inner capsule sizes, (**a**) 1.7 μm, (**b**) 7.5 μm, (**c**) 10.0 μm. **d–f** Fluorescence images of the emulsions with varied inner droplet volume fractions and SEM images of their derived microstructures, (**d**) 80%, (**e**) 70%, (**f**) 65%. **g–i** SEM images of the microstructures from varied inner droplet volume fractions by a pre-coordination process (for 4 h), (**g**) 80%, (**h**) 70%, (**i**) 65%. [(**a₁–f₁**) Fluorescence images, (**a₂–c₂**) low-magnification observation, (**a₃–c₃**, **d₂–f₂**, **g₁–i₁**) a single broken particle, (**a₄–c₄**, **d₃–f₃**, **g₂–i₂**) high-magnified observation of the interior structure.] The insets show 3D models of the corresponding architecture. **j** Schematic illustration of the structural evolution of multi-compartmental MOF microreactors with various interior microstructure.

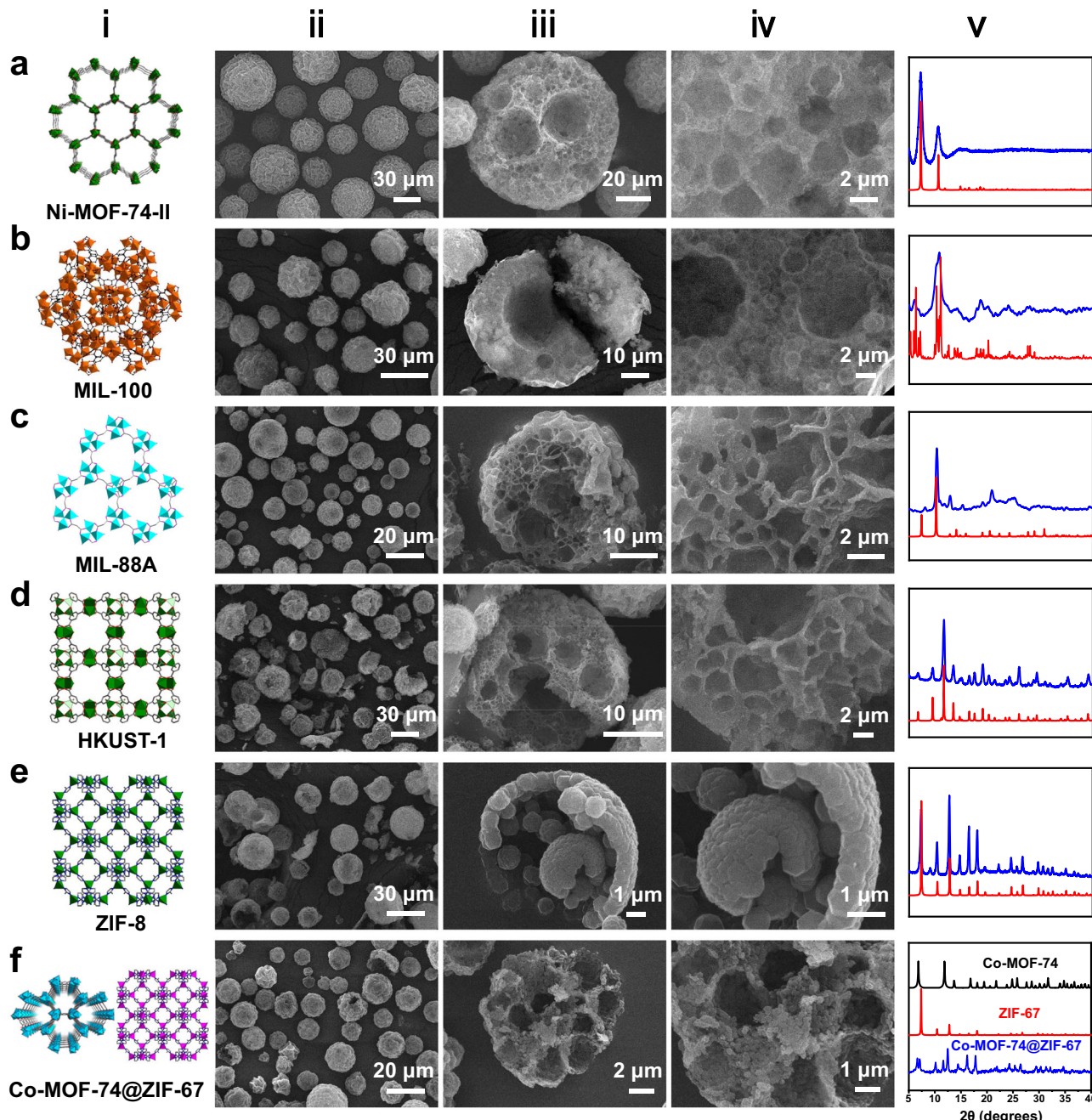

**Fig. 4 | Series of multi-compartmental MOF microreactors derived from Pickering double emulsions. a** Ni-MOF-74-II, **b** MIL-100, **c** MIL-88A, **d** HKUST-1, **e** ZIF-8, and **f** Co-MOF-74@ZIF-67. (i) Crystal structures, (ii) low-magnified SEM images, (iii) a single broken particle, (iv) high-magnified interior structure, (v) the experimental (blue) and simulated (red and black) XRD patterns.

obtained under similar procedures, implying the possibility and ease of industrial transposition of this methodology.

## Encapsulation ability

Benefiting from the integrated ability of both Pickering double emulsion and crystalline microporous MOF structures, such a unique architecture is expected to be utilized as an ideal reservoir for physically confining various functional species. To examine this, we then studied the encapsulation ability of our multi-compartmental MOF microparticles towards a range of guest species, including hydrophilic enzymes, like Candida antarctica lipase B (CALB, $3.0 \times 4.0 \times 5.0 \, nm^3$), glucose oxidase (GOx, $7.0 \times 5.5 \times 8.0 \, nm^3$) and horseradish peroxidase (HRP, $4.0 \times 4.4 \times 6.8 \, nm^3$), and hydrophobic molecular compounds,

like Grubb' catalyst (molecular size is about $1.45 \times 1.11 \times 0.63 \, nm^3$) and Fe-porphyrin (denote as TPP(Fe), molecular size is about $1.74 \times 1.74 \times 0.43 \, nm^3$) (Fig. 5a and Supplementary Figs. 22a and b). Positional loading of these enzymes was realized by adding a desired quantity of enzymes into the aqueous phase before the secondary emulsification process. Impressively, all the proteins could be successfully confined into the MOF microparticles with a high encapsulation efficiency (>95%, Fig. 5b and Supplementary Table 2). As visualized by a time-dependent CLSM records (Fig. 5c and Supplementary Fig. 23), FITC-labeled CALB or Rhodamine B-labeled GOx (or HRP) enzymes were observed homogeneously distributing throughout the whole microreactors, and almost no fluorescence signals were found in their polar surroundings with time prolonged, suggesting the

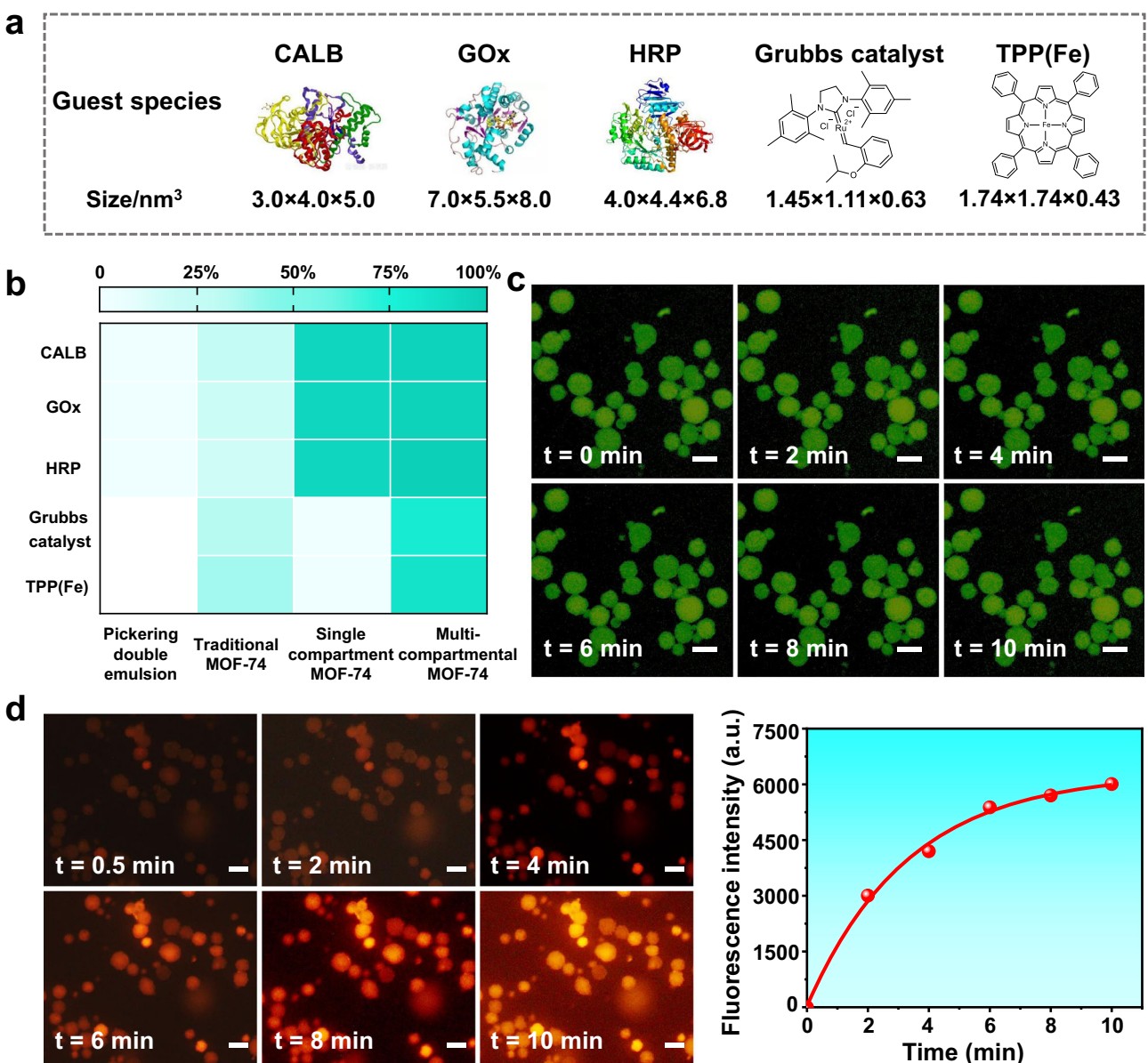

**Fig. 5 | Encapsulation ability of the multi-compartmental MOF microreactors.**
**a** Structural illustration of the enzyme or molecular probes. **b** Comparisons of the encapsulation efficiency for hydrophilic CALB, GOx, HRP and hydrophobic Grubbs, TPP(Fe) in different systems. **c** Confinement behavior of large-sized enzymes in polar surrounding (ethanol), as visualized by fluorescence microscopy of the FITC-labeled CALB. **d** Fluorescence confocal microscopy images and their intensity as a function of time for the diffusion of Nile Red (8.0 μM) into the MOF-74 microreactor, scale bar = 30 μm.

good encapsulation ability. To benchmark the superior encapsulation efficiency, other contrastive systems were also performed. For the pure double emulsion system, the droplets without an MOF skeleton will be broken in polar solvent (Supplementary Fig. 24), leading to very poor encapsulation efficiency (<5%). As for the traditional solvothermal system (Supplementary Fig. 25), a limited loading efficiency (<30%) was found due to its intrinsic drawbacks. Encouragingly, the multi-compartmental MOF microreactors also demonstrated a significantly higher loading efficiency for rigid molecular catalysts (by pre-dissolving them in the inner oil droplets) than other systems. For example, the encapsulation efficiency of Grubb' catalyst and TPP(Fe) in the multi-compartmental MOF microparticles was nearly 80%, which is ~4-fold higher than traditional solvothermal synthesis system. Notably, although a single compartment MOF structure derived from simple water-in-oil Pickering droplet could efficiently encapsulate hydrophilic enzymes (Supplementary Fig. 26), it is not adopted to

hydrophobic species. These results sufficiently confirmed the excellent spatial confinement ability of our multi-compartmental MOF architectures towards both hydrophilic and hydrophobic species, thereby the spatiotemporal programming of the different functionalities might be achieved after simultaneously confining them into an integrated system.

Next, we sought to investigate the accessibility of our MOF microreactors to smaller molecules, which is vital for their potential catalysis applications. In this regard, we selected a fluorescent Nile Red, with a theoretical molecular size of $1.43 \times 0.575 \times 0.519$ nm$^3$ (Supplementary Fig. 22c), as an effective probe to assess the molecular permeability. As shown in Fig. 5d, an increasing fluorescent intensity throughout the MOF microparticles was clearly observed with time going on, indicating the steady molecular transportation of Nile Red from the external surrounding to the particle interior space. The smooth molecular diffusion could be ascribed to the non-

interpenetrating crystallites and large apertures (-15 Å) of MOF-74 materials.

## Chemo-enzymatic cascade catalysis

Equipped with multi-compartmental structure, versatility, and integrated functionality, our developed MOF architectures can be employed to spatialize different catalytic species in one system for operating multi-step cascade reactions. As a proof of concept, a two-step chemoenzymatic reaction, based on a well-known Grubb' catalyst and CALB lipase, was first conducted (Fig. 6a)[64,65]. These two distinct catalytic species were encapsulated simultaneously within a single MOF microreactor (MOF-74 as an example) by preloading them in the inner oil and water compartments, yielding Grubbs/CALB@MOF catalyst. SEM imaging showed that the morphology and structure were not disturbed with incorporating these species (Supplementary Fig. 27). Confocal fluorescence microscopy observations showed the existence of CALB enzyme within the MOF microreactor, which was further verified by FT-IR characterization (Fig. 6b). The characteristic band ascribing to amide-I stretch at 1651 $cm^{-1}$ was clearly identified in the Grubbs/CALB@MOF, while the adsorption peaks of v $_{Ru=N}$ that related with the ligand of Grubbs' catalyst appeared around 748 $cm^{-1}$ (Supplementary Fig. 28). Moreover, the FT-IR spectra of Grubbs/CALB@MOF also revealed a peak

shift compared with freestanding CALB enzyme (amide-I stretch, blue shift from 1638 $cm^{-1}$), which might be attributed to the existed enzyme-MOF interaction that originates from the coordination of metal ions and carbonyl groups of enzymes. EDX elemental mappings showed uniform distribution of Ru, N, P, S over the whole Grubbs/CALB@MOF, further confirming the coexistence of enzyme and Grubbs' moieties in the complex cellular architectures (Fig. 6c).

Given the above results, we then examined the catalytic performance of Grubbs/CALB@MOF for the one-pot ring-closing metathesis (RCM)/transesterification cascade reaction using 1, 6-heptadien-4-ol as a model substrate (Fig. 6a). Separated catalysts, Grubbs@MOF and CALB@MOF, were also performed to determine the reaction route is I (RCM first and transesterification later) or II (transesterification first and RCM later). As shown in Supplementary Fig. 29, only RCM or transesterification reaction occurred in the presence of Grubbs@MOF or CALB@MOF, and the RCM step proceeded much faster than transesterification, indicating that the cascade reaction mainly happened by route I. As expected, the integrated Grubbs/CALB@MOF could efficiently drive the cascade reaction, and achieved a 49.0% conversion with 74.6% selectivity of final 3-cyclopentenyl acetate (CA) product within 90 min (Figs. 6d$_1$ and 6d$_2$). To highlight as well as clarify this activity, several contrast experiments, including the homogeneous counterparts (Grubbs/CALB), mechanical mixture of the separated catalysts (Grubbs@MOF/CALB@MOF), traditional solvothermal immobilization system (Grubbs/CALB@traditional MOF) and mesoporous SBA-15 immobilized synthetic system (Grubbs/CALB@SBA-15) were also performed. Obviously, Grubbs/CALB@MOF exhibited much better activity and selectivity than all of the other four systems. As shown in Figs. 6d$_1$ and 6d$_2$, only 12.5, 25.6, 17.0% conversions and 16.3, 60.9, 49% selectivities were obtained for the free Grubbs/CALB, Grubbs@MOF/CALB@MOF and Grubbs/CALB@SBA-15, while almost no reaction occurred for the Grubbs/CALB@traditional MOF under identical reaction conditions. To better evaluate such distinct catalytic results, catalysis efficiency (CE) of these systems was also estimated based on the reaction profiles. As shown in Fig. 6d$_3$, the integrated Grubbs/CALB@MOF affords a CE value of 21.5 mol $mol^{-1} h^{-1}$, which is 5.81, 2.24 and 4.06-fold higher than that of free Grubbs/CALB system (3.7 mol $mol^{-1} h^{-1}$), Grubbs@MOF/CALB@MOF system (9.6 mol $mol^{-1} h^{-1}$) and Grubbs/CALB@SBA-15 (5.3 mol $mol^{-1} h^{-1}$), respectively. The relatively inferior catalytic efficiency of the homogeneous mixture can be ascribed to the intrinsic incompatibility of the two different world catalytic species, which is

verified by a drastically reduced activity after the addition of CALB enzyme in the homogeneous Grubb' reaction solution (Supplementary Fig. 30). In this respect, the spatially confined MOF systems will benefit the overall activity, since the segregated compartments might shield the incompatible chemo- and bio-catalytic species from each other, thus preventing their mutual inactivation. More importantly, the positional assembly of these species in a single multi-compartment system also significantly promote the cascade efficiency of Grubbs/CALB@MOF depends on the contributions of directed mass transportation between compartments. Within a Grubbs/CALB@MOF microreactor, the intermediate product 3-cyclopenten-1-ol produced from the RCM reaction in the inner compartments, and then was freely transferred into the nearby MOF skeletons to trigger the enzymatic transesterification reaction immediately. During this process, the close confinement and dense packing compartments enable a high local concentration and effective transport of intermediates at the enzymes like in living cells, thereby promoting substrate channelling and enhancing the cascade efficiency in one pot[66]. On the contrary, for the non-organized Grubbs@MOF/CALB@MOF system, it suffers from lower local concentration of the intermediates due to the inevitable dilution after diffusion into the bulk solution, thus leading to a slower catalytic efficiency. On the other hand, the mild synthesis conditions and hierarchical porous structures of Grubbs/CALB@MOF are also important factors for the enhanced catalytic performance. When the catalytic species were co-immobilized by a traditional solvothermal synthesis way (temperature is 100 °C), significant deactivation would happen, resulting in deteriorative catalysis efficiency. As for the immobilized synthetic system Grubbs/CALB@SBA-15, the relatively poor activity might be related with its low loading efficiency and hydrophilic channel properties, which is unfavorable for the catalytic activity of both Grubb' catalyst and CALB species. Moreover, we also studied the impacts of reaction temperature on the catalysis efficiency to optimize the reaction conditions. As shown in Supplementary Fig. 31, the reaction temperature shows a significant influence on the catalytic performance of cascade reaction. Along with the increasing of reaction temperature from 25 to 45 and 70 °C, the overall conversion gradually increased from 21.0 to 48.9 and 84.2%, while the selectivity of CA drastically improved from 19.8 to 74.6 and 86.3%. Meanwhile, the corresponding CE was calculated to be 6.2, 21.5 and 32.7 mol $mol^{-1} h^{-1}$. Although better activity was achieved at a higher temperature of 70 °C, we decided to carry out the reactions at 45 °C, since the good stability of enzymes under mild working conditions.

In addition, thanks to the in situ confined pathway of our method, the local catalyst loading and mass ratio of Grubbs/CALB could be easily regulated on demand to optimize the cascade reaction. As show in Figs. 6e$_1$ and 6e$_2$, upon increasing the loading of CALB from 2.0 to 8.0 mg $g^{-1}$ while keeping the loading of Grubb's catalyst at 25 mg $g^{-1}$, the overall conversion gradually increased from 37.5% to 56%, while the selectivity of CA drastically improved from 36.2% to 94.6%. Meanwhile, the corresponding CE was calculated to be 15.6, 21.5 and 24.6 mol $mol^{-1} h^{-1}$ (Fig. 6e$_3$). These results are easy to be understood. Since the enzymatic step is much slower than the initial step motivated by Grubb' catalyst, the conversion of the cascade reaction is mainly dominated by the enzyme, thus leading to gradual increase of conversions with varying enzyme contents. The intermediates will be more efficiently converted to final CA product if the enzyme content is increased, resulting in coordinating reaction kinetics with significantly enhanced selectivity. After optimizing the mass ratio of Grubbs/CALB, we further increased the quantity of these species to 50 and 8 mg $g^{-1}$, so as to get an impressive conversion and selectivity as high as 97.6 and 98.1% respectively. These above results indicate that our strategy allows the complex cascade reactions to be optimized in a rational and controlled manner. Furthermore, our integrated catalyst Grubbs/CALB@MOF could be expanded to the chemo-enzymatic cascade reaction of other substrates. As shown in Supplementary Fig. 32, for

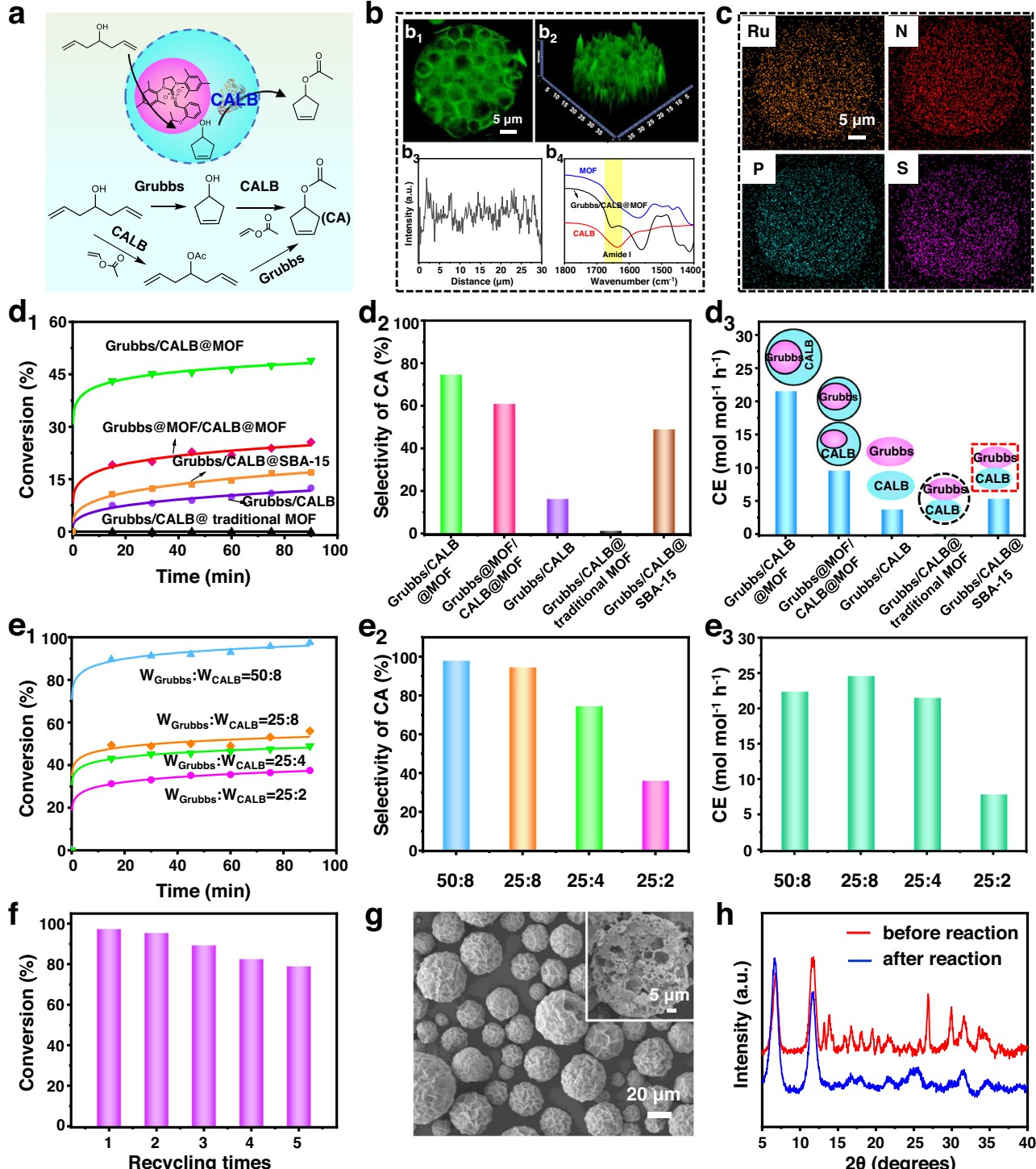

**Fig. 6 | Chemo-enzymatic cascade catalysis in Grubb' catalyst/CALB lipase driven olefin metathesis/transesterification reaction. a** Schematic showing the one-pot cascade situation. **b** Confocal fluorescence microscopy and FT-IR of Grubbs/CALB@MOF. 2D (**b₁**) or 2.5D (**b₂**) image of FITC-Dextran-labeled CALB within the multi-compartmental microreactor and its corresponding fluorescence intensity profile (**b₃**), FT-IR spectra (**b₄**). **c** EDX mappings of Grubbs/CALB@MOF for the characteristic elements, including Ru, N, P and S. **d** Catalytic performance of the cascade reaction in different systems, including Grubbs/CALB@MOF, Grubbs/ CALB, Grubbs@MOF/CALB@MOF, Grubbs/CALB@traditional MOF and Grubbs/ CALB@SBA-15. Kinetic plots (**d₁**), selectivity of aimed CA product (**d₂**), and calculated catalysis efficiency (**d₃**). **e** Catalytic results of Grubbs/CALB@MOF with different weight ratios of enzyme and molecular catalyst; Kinetic plots (**e₁**), selectivity of aimed CA product (**e₂**), and calculated catalysis efficiency (**e₃**). **f** Recyclability of Grubbs/CALB@MOF. **g** SEM images of Grubbs/CALB@MOF after reaction. **h** XRD patterns before and after reaction.

substrate 4-methyl-1,6-heptadien-4-ol (Reaction I), a conversion of 80.1% together with a selectivity of 56.3% were obtained in the one-pot RCM/transesterification cascade reaction. Meanwhile, the solid Grubbs/CALB@MOF was applied to the RCM/hydrolysis cascade reaction with diethyl diallylmalonate as a substrate (Reaction II), achieving a conversion of 98.0% with a selectivity of 29.0%. This inferior selectivity might be ascribed to the biphasic catalysis system, which lead to much lower catalytic activity of the enzymatic hydrolysis step. These extended catalytic results further demonstrated the generic applicability of our Grubbs/CALB@MOF in chemo-enzymatic cascade reactions.

To verify the stability and recyclability of our solid Grubbs/CALB@MOF catalyst, a filtration experiment, and recycling tests were also conducted. As shown in Supplementary Fig. 33a, after a reaction time of 20 min, the solid catalyst was filtered from the system and the filtrate left was allowed to further react for another 90 min. It was found that the conversion of the filtrate remains almost constant at 43.0%, while the system always in the presence of the Grubbs/CALB@MOF catalyst gave more than 48.0% conversion. This result indicates that the catalytic species were well confined and didn't leach out during the reaction, which is further verified by the UV–Vis analysis of the filtrate solution (Supplementary Fig. 33b). Moreover, our solid Grubbs/CALB@MOF catalyst could be easily recycled at least five times with remaining 79.3% catalytic activity (Fig. 6f). After reaction, the morphology and well-defined multi-compartmental microstructure are essentially unchanged, except for a slight degradation of the MOF structure, as confirmed by the SEM images and XRD patterns (Fig. 6g, h). The decreased activity might be relevant to the deterioration of Grubbs molecular catalyst caused by their decomposition or dimerization under reaction conditions[67].

Encouraged by the above results, we then conducted the second chemo-enzymatic cascade reaction, GOx/Fe-porphyrin driven oxidation of ABTS [ABTS = 2,2'-azino-bis (3-ethylbenzothiazoline-6-sulfonic acid), molecular size is about $1.70 \times 0.80 \times 0.18 \ nm^3$] (Supplementary Figs. 34a and 35), to check the generality of our method. This reaction involves GOx-mediated oxidation of glucose to form $H_2O_2$ followed by TPP(Fe) catalyzed ABTS oxidation to produce the fluorescent $ABTS^{+}\bullet$ product (Supplementary Fig. 34b)[68]. These catalytic species were co-confined within the multi-compartmental MOF microreactors through a similar preloading way, and verified by a confocal fluorescence microscopy observation (Supplementary Fig. 34c) and FT-IR characterization (Supplementary Fig. 36). As expected, the yielded GOx/TPP(Fe)@MOF could effectively drive the cascade oxidation reaction, producing $ABTS^{+}\bullet$ product, as demonstrated by the appearance of absorption peak at 415 nm (Supplementary Fig. 34d). In comparison, neither separated catalysts, GOx@MOF or TPP(Fe)@MOF, could realize the reaction under similar conditions, indicating the absolutely necessary of both catalytic species. The time-dependent monitoring of $ABTS^{+}\bullet$ product by UV–Vis measurement shows a smooth increasing of the absorbance along with time, resulting in a blue color in solutions (Supplementary Fig. 34e). By contrast, much lower catalytic performance was observed in the mechanically mixed system GOx@MOF/TPP(Fe)@MOF or homogeneous counterpart GOx/TPP(Fe). The enhanced catalytic activity might be ascribed to the fact that GOx within the confined space of GOx/TPP(Fe)@MOF yields a high local concentration of $H_2O_2$ in the vicinity of TPP(Fe), thus leading to an effective cascade. These results again provide remarkable proof for the adaptability of our method in dealing with chemo-enzymatic cascade catalysis.

## Discussion

In summary, we report a Pickering double emulsion-directed interfacial synthesis method for efficiently constructing multi-compartmental crystalline MOF microreactors, which can be utilized as a powerful platform for integrating orthogonal catalytic species for multistep chemo-enzymatic cascade reactions. Compared with previously reported methods, this developed strategy exhibits several attractive features. Firstly, the chosen Pickering double emulsion can provide reliable multiple liquid–liquid interfaces for the self-completing growth of dense MOF layers, thus enabling the fabricated microreactors with tailor-made architectures and selective permeability. Secondly, the elaborate control over the interior architecture of the MOF microreactors, in particular the size and packing degree of inner compartments as well as their well-defined hollow structures, could be unprecedentedly realized, providing sufficient freedom for customizing the material's structure and functional properties on demand. Thirdly, such a strategy affords great flexibility, extendibility and scalability, making it applicable for large-scale production of a variety of MOF structures. More importantly, spatially controlled organization of incompatible functionalities inside the tailored synthetic compartments, including hydrophilic enzyme and hydrophobic molecular catalyst, could be conveniently realized in a bottom-up manner, forming biomimetic functional systems for operating concurrent chemo-enzymatic reactions in a one-pot fashion. As exemplified by the Grubb' catalyst/CALB lipase driven olefin metathesis/transesterification reaction and glucose oxidase/Fe-porphyrin catalyzed oxidation reaction, the power of our designed multi-compartmental microreactor was definitely demonstrated by the significantly enhanced catalytic efficiency (2.24–5.81 folds) in comparison to the homogeneous counterparts or physical mixture of individual analogues. The restrained mutual inactivation of enzyme and molecular catalyst within the compartments, together with the promoted substrate channelling effects, might be attributed to the enhanced cascade efficiency. We believe that this rational synthesis approach offers opportunities for the design and development of complex multi-compartmental cellular models with spatially organized functions, taking a step forward the artificial cells that is closer to living systems. Furthermore, in view of the fact that MOFs are extensively applied in the fields of drug delivery, gas sorption/separation, energy storage and release, and biomedicine et al., the developed MOF superstructures are also expected to serve as an alternative platform for the above applications.

## Methods

### Preparation of O/W/O Pickering double emulsions

Firstly, 1.0 mL of *n*-octanol solution that containing a desired amount of organic ligand (2,5-dihydroxyterephthalic acid, 0.012 mmol) and tributylamine (30 μL) was added into an aqueous phase that containing 5.0 wt% hydrophilic emulsifier (with respect to oil phase). The biphasic mixture was homogenized for 2 min at 400 W using a SCIENTZ-IID homogenizer, forming a primary O/W single emulsion. Then, this emulsion was collected through centrifugation at 5000 rpm, and the bottom excess water phase was removed, resulting a dense packing high internal phase emulsion (HIPE). Subsequently, an aqueous solution of metal salts [$Ni(NO_3)_2 \cdot 6H_2O$ (0.69 mmol) dissolved in 200 μL water, 24 mg PVA was added at the same time] was added into the above emulsion, followed by mixing it with 5.0 mL of *n*-octanol that containing 40 mg hydrophobic silica emulsifier. After vigorous hand-shaking, a stable O/W/O double emulsion was obtained. For loading with enzyme or molecular species, the procedures were similar except that a desired amount of hydrophobic molecular compounds or hydrophilic enzymes were pre-dissolved in the inner oil or aqueous phases.

### General procedure for the fabrication of multi-compartmental MOF-74 microreactors

The above resulting Pickering double emulsion was dispersed into a *n*-octanol solution of organic ligand (50 mL, 0.35 mmol 2,5-dihydroxyterephthalic acid, 200 mg tributylamine), and rotated on a rolling apparatus at 40 °C overnight. Then, the upper oil phase was

removed and the solid products were isolated by centrifugation, washed with ethanol for several times, and dried under vacuum. The multi-compartmental microreactor hosting with hydrophobic molecular catalyst and hydrophilic enzyme was prepared similarly except that molecular catalyst/enzyme-loaded Pickering droplets were used.

## Data availability

The data for Figs. 2–6 generated in this study are provided in the Supplementary Information/Source Data file. Source data are provided with this paper. All data is available from the corresponding author upon request. Source data are provided with this paper.

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

## Acknowledgements

This work is supported by the Natural Science Foundation of China (21925203 and 22072078), the Outstanding Youth Fund of Shanxi Province (202103021222003), and the National Key Research and Development Program of China (2021YFC2101900).

## Author contributions

H.Q.Y. and X.M.Z. conceived and supervised the project; D.P.T. and R.P.H. designed and carried out experiments; Y.W.W. and L.F.L. conducted XRD measurements; H.S. performed the DFT computations. H.C.L. contributed in designing and discussing the experiments. D.P.T., X.M.Z., and H.Q.Y. wrote the paper. All authors contributed to the manuscript.

## Competing interests

The authors declare no competing interests.
