## [Peer Review File · Nature Communications]

Multi-Compartmental MOF Microreactors Derived from Pickering Double Emulsions for Chemo-Enzymatic Cascade CatalysisReviewers' Comments:

Reviewer #1:

Remarks to the Author:

The manuscript submitted by Tian et al. demonstrated an interesting and general Pickering double emulsions-based method for building multi-compartmental MOF microreactors that can be skillfully utilized in chemo-enzymatic cascade catalysis. I noticed that this group have done in depth work in "Pickering" system. In this particular work, the authors integrate the precise manipulation capability of Pickering double emulsions, thus enabling the fabricated microreactor with tailor-made interior multi-compartment architectures. In addition, this strategy was confirmed to be flexible, extendible and scalable for the production of a variety of MOF structures. Furthermore, in the application aspect, simultaneous organization of incompatible hydrophilic enzyme and hydrophobic molecular catalyst inside the tailored synthetic compartments was demonstrated to be reliable for operating concurrent chemo-enzymatic reactions. The significant enhancements in catalytic efficiencies further confirmed the superior advantage of the ingenious design. The manuscript is wonderfully outlined and the results are discussed by thoroughly performed experiments. Therefore, I recommend the manuscript to publish in Nature Communication after minor revisions listed below.

(1) The authors should clarify further how the present system is superior compared to previous reports and how they came up with this specific design?

(2) The f3 in Figure 3 and Figure 4d (iv), the graphic quality is too poor to identify the multi-compartment architecture. I suggest the authors to provide properly visible and representative SEM images.

(3) The approach can be applied for the synthesis of hybrid MOF systems by using different organic linkers. How to prove that organic ligands do not migrate between the internal and external oil phases during synthesis? For figure 4e (iii) and (iv), the inner structures of ZIF-8 microreactor seem similar to the building blocks of MOF external skeleton. Please clarify these in the revised manuscript.

(4) Can the authors comment on the recyclability or reusability of the present system? How does the performance of the authors' materials compare to other synthetic systems where the enzymes or molecular catalysts are immobilized in some way?

Reviewer #2:

Remarks to the Author:

The authors present a novel concept to synthesize MOF-walled microreactors for cascade reactions by employing a Pickering double emulsion "oil-on-water-on-oil" route. The basic idea is to prepare an emulsion@emulsion system in which MOF "walls" are grown inside an emulsion, which then leads to a microreactor system consisting of small, hollow MOF spheres inside a larger MOF sphere. By this route, both the smaller and the larger cavities can separately (i.e. consecutively) be "filled by different catalysts with the aim to perform cascade catalysis. The authors show that they are able to finetune and to control the size and density of the inner and outer microreactor by a multitude of different methods.

This concept of synthesizing such kind of MOF@MOF materials is very innovative since it allows for a precise incorporation of each MOF cavity with different catalysts, which are spatially separated and thus the reactions that are catalyzed by the spatially separated catalysts should not interfere negatively or inhibit each other during the desired catalytic cascade reaction.

In principle, I am quite fascinated by the beauty of the concept and in favor of the manuscript; nevertheless, there are some points regarding the significance of the catalytic data and the selection of the model reactions for demonstration of the system's performance which should be clarified or substantiated before a decision on accepting the paper for publication can be made:

- The kinetics of the formation of the MOF walls and the control over their thickness are somewhat

unclear, could the authors be more specific about this?

- The 1st catalytic cascade reaction, i.e. the ring closing metathesis (with a Grubbs catalyst in the small reactors)/transesterification (with a CALB enzyme in the larger reactors) is, as the authors pointed out, kinetically controlled: the 1st reaction is much faster than the 2nd one, which can be adjusted by increasing the concentration of the CALB catalyst. Nevertheless, this does not seem to be a unique feature of the authors' catalyst system; after all, the fully homogeneous Grubbs/CALB system does not perform significantly worse and it has not been optimized. Thus, it is not convincing to state that there is an "intrinsic incompatibility" of these two catalysts in the homogeneous phase (l. 413, p.20); this is not at all shown in the referenced Supp. Fig. 29 of the SI on p. S33. The initially low conversion/selectivity of the product could be interpreted in a way that vinyl acetate (VA) also undergoes metathesis with another VA molecule or with 1,6-heptadiene-4-ol, which could mean that the selectivity of the reaction is not controlled by the MOF@MOF microreactor system. Only an increase of the CALB conc. can lead to a faster decrease of the VA conc. and thereby suppress side reactions. In this regard, one could pose a provocative question: why does it require such a sophisticated [Catalyst 1@MOF]@[Catalyst 2@MOF] composite if both reactions can proceed simultaneously even if both catalysts can exist in one phase? What is the key unique feature of the system – beyond a somewhat faster and more efficient reaction?

- Does an alteration of the substrates (e.g. by polarity) have an influence on the reaction kinetics (e.g. by different diffusion rates)? Do the surfactants of the SiO₂ NPs play a role in the interaction with the substrates?

- Fig. 6f/recycling: the Fig. shows the conversion steadily decreasing with each run, which, together with the XRDs in Fig. 6h, points to a degradation of the MOF structure (and possible catalyst leaching due to a partial collapse of the MOF walls), which in turn speaks against a good stability (as stated on p.22, l.458/459). Can the authors comment on this discrepancy? Has the biphasic system been stirred during the reaction? Is there a difference in MOF stability after the catalytic reaction for different MOF types?

Reviewer #3:

Remarks to the Author:

This study presented a nice method of constructing multi-compartmental MOF microreactors from pickering double emulsions for chemo-enzymatic cascade catalysis. The structure of the multi-compartmental MOF microreactors is novel and important for the high efficiency of chemo-enzymatic cascade catalysis. The Grubb' catalyst/CALB and GOx/Fe systems encapsulated in the microreactors exhibited 2.24-5.81 fold enhanced catalytic efficiency. And the authors found that the enhanced efficiency is a result of spatial confinement of two incompatible catalysts and the increased transportation efficiency of the reaction intermediate. Overall, this is a well-organized manuscript which can be accepted after addressing minor points:

1) When discussing the substrate channeling effect of the reaction intermediate between CALB and the Grubb' catalyst, the following reference would be helpful: Nano Letters 2022, 22, 5029-5036

2) In the cascade reaction, the Grubb' catalyst and CALB may work at different optimal temperatures. Did the authors try to investigate the effect of reaction temperature of this cascade reaction?

3) Page 22, "Since the enzymatic step is much slower than the initial step motivated by Grubb' catalyst, the conversion of the cascade reaction is mainly dominated by the Grubb' catalyst, thus leading to gradual increase of conversions with varying enzyme contents." should be "...is mainly dominated by enzyme..."

Reviewer 1

The manuscript submitted by Tian et al. demonstrated an interesting and general Pickering double emulsions-based method for building multi-compartmental MOF microreactors that can be skillfully utilized in chemo-enzymatic cascade catalysis. I noticed that this group have done in depth work in “Pickering” system. In this particular work, the authors integrate the precise manipulation capability of Pickering double emulsions, thus enabling the fabricated microreactor with tailor-made interior multi-compartment architectures. In addition, this strategy was confirmed to be flexible, extendible and scalable for the production of a variety of MOF structures. Furthermore, in the application aspect, simultaneous organization of incompatible hydrophilic enzyme and hydrophobic molecular catalyst inside the tailored synthetic compartments was demonstrated to be reliable for operating concurrent chemo-enzymatic reactions. The significant enhancements in catalytic efficiencies further confirmed the superior advantage of the ingenious design. The manuscript is wonderfully outlined and the results are discussed by thoroughly performed experiments. Therefore, I recommend the manuscript to publish in Nature Communication after minor revisions listed below.

Response: We are very grateful to your positive and supportive comments for this work. According to your advice, we have improved our manuscript, as listed below.

1. The authors should clarify further how the present system is superior compared to previous reports and how they came up with this specific design?

Response: In our opinion, the superiority of our designed multicompartmental MOF system compared to previous reports is mainly expressed in the following aspects. Firstly, different from the reported multi-compartment systems (like lipid vesicles, hydrogels, polymersomes, porous silica or carbon et al), the building MOF matrix not only possesses good mechanical strength and chemical robustness, but also shows selective permeability, which is very important for their potential applications in spatial confinement of active species within the compartments. Meanwhile, compared with the reported single compartment MOF structures, this study unprecedentedly achieved the construction of hierarchically multi-compartmental MOF microreactors, of which the interior architectures can even be finely engineered. Furthermore, from the application perspective, the emerging concurrent chemo-enzymatic cascades has not been implemented in a single MOF microreactor due to their mutual incompatibility stemming and distinct physicochemical properties. This proposed Pickering double emulsion-based interfacial synthesis method can readily realize the spatial confinement of incompatible chemo- and bio-catalytic species, including hydrophobic organometallic catalysts and hydrophilic enzymes. In fact, we had given such related statements as “*Despite these intriguing advances, severe obstacles are still standing in the way of moving closer to a truly applicable cell-like microsystem. On the one hand, most of the reported multi-compartment systems involve soft matter in nature, thereby suffering from the shortfalls of poor mechanical strength and chemical robustness*”

required by practical applications. On the other hand, selective permeability, which ensures biomimetic properties like membrane gating, is far less achieved till now, causing inevitable difficulty in spatial confinement of active species or uncontrollable molecular transport within the compartments.”, “However, these reported microreactors are basically limited to single compartment structures, thus resulting in lack of control over the spatial organization of different catalytic actives and inapplicability for the accommodation of incompatible catalytic transformations in one vessel. Especially, the emerging concurrent chemo-enzymatic cascades, which offer complementary means to unify the advantages of synthetic catalysts with natural enzymes, has not been implemented in a single MOF microreactor due to their mutual incompatibility stemming and distinct physicochemical properties^{47,48}.” and “we develop a facile and efficient Pickering double emulsion-based interfacial synthesis method to generate hierarchically multi-compartmental MOF microreactors that can readily realize the spatial confinement of incompatible chemo- and bio-catalytic species for operating effective chemo-enzymatic cascade reactions. In sharp contrast to previous reports⁴⁹⁻⁵³, this approach involves the utilization of stable Pickering double emulsions as a directing template for the growth of crystalline MOF structures, which seems to be a good choice, since such a type of emulsion provides not only large liquid-liquid interfacial areas for self-completing formation of dense MOF layers, but also independent space to introduce diverse functional species^{54,55}.”

2. The f_3 in Figure 3 and Figure 4d (iv), the graphic quality is too poor to identify the multi-compartment architecture. I suggest the authors to provide properly visible and representative SEM images.

Response: According to your suggestion, we have carefully improved the graphic quality of f_3 in Figure 3 and Figure 4d (iii, iv) in our revised manuscript, and now they are clearly visible (as shown below).

Figure A (Figs. f_2 and f_3 in the revised manuscript). SEM images of the microstructures from an inner droplet fraction of 65%.

Figure B (Figs. 4d iii and iv in the revised manuscript). A single broken particle and high-magnified interior structure of HKUST-1 microreactor.

3. The approach can be applied for the synthesis of hybrid MOF systems by using different organic linkers. How to prove that organic ligands do not migrate between the internal and external oil phases during synthesis? For figure 4e (iii) and (iv), the inner structures of ZIF-8 microreactor seem similar to the building blocks of MOF external skeleton. Please clarify these in the revised manuscript.

Response: The organic ligands dissolved in the internal and external phases would not migrate between each other. Firstly, the Pickering double emulsions are very stable, providing a protection towards the dispersed droplets, and the diffusion resistance of organic ligand molecules that initially dissolved in the inner oil droplets across aqueous phase to outer oil phase is very strong. Moreover, the quickly formed MOF skeleton will also restrict the diffusion of organic ligands. In fact, to explore the excellent stability of the Pickering double emulsions and visualize the possible migration of confined molecules between the internal and external oil phases, we have performed a series of fluorescence dyeing experiments. As shown in Fig. 2c, when a hydrophobic probe molecule, Nile Red, was preloaded within the internal oil droplets, its red fluorescence signals always existed within the internal droplets over the whole monitored time, suggesting that the dye molecules did not diffuse through the aqueous phase to enter the external oil surrounding. If this dye molecule was initially dissolved in the external oil phase, almost no fluorescence signal was detected within the internal droplets (Supplementary Fig. 6). Similarly, as the hydrophilic FITC-Dextran was preloaded, green fluorescence signal was only observed within the aqueous zone (Supplementary Fig. 7). These results sufficiently demonstrate the positional compartmentalization ability of the double emulsions and limited molecular migration between different phases. Thus, this method can also be applied for the synthesis of hybrid MOF systems by using different organic linkers. Taking Co-MOF-74@ZIF-67 as an example, H₄DOBDC and 2-methylimidazole are respectively dissolved in the inner oil droplets and outer oil continuous phase, so tend to coordinate with the metal ions at the internal and external interfaces, forming hybrid MOF composites. The XRD pattern of resulting materials matches well with the simulated Co-MOF-74 and ZIF-67 respectively [Fig. 4f (v)], verifying the formation of targeted MOF crystal structures.

As for the multi-compartmental ZIF-8 microreactor, the solubility of its utilized ligand 2-methylimidazole is quite different from other organic ligands, which can be

soluble in both aqueous and organic phases. Therefore, the forming interior structure of ZIF-8 microreactor is distinct from other types of materials. As illustrated in Fig. 4e (iii) of the manuscript, the SEM image clearly shows that the inner sub-compartments and external skeleton of ZIF-8 microreactor were both assembled from rhombic nanocrystals. According to your suggestion, we have improved the images of high-magnified interior structure in Figure 4e (iv) of our revised manuscript, also shown as below. From these images, both of the outer MOF shell and interior structure can be clearly observed. Meanwhile, we have clarified the related description as “*Specifically, the interior structure of ZIF-8 microreactor is distinct from other types of MOF materials [Fig. 4e (iii, iv)], since the solubility of its utilized 2-methylimidazole is quite different from other organic ligands, which could be dissolved in both aqueous and organic phases.*”

Figure C (Figs. 4e iii and iv in the revised manuscript). *A single broken particle and high-magnified interior structure of ZIF-8 microreactor.*

4. Can the authors comment on the recyclability or reusability of the present system? How does the performance of the authors' materials compare to other synthetic systems where the enzymes or molecular catalysts are immobilized in some way?

Response: For recycling tests shown in Fig. 6f, we can see that the conversion steadily decreased with each run. To clarify the causes of this result, SEM observations together with XRD characterizations for the sample after reaction were conducted. Firstly, from the XRD patterns, we can see that most of the peaks are virtually unchanged, and the SEM images also confirm the good retain of morphology and multi-compartmental microstructure after recycling. These results indicating that the structure degradation is relatively slight, which is not enough to lead catalyst leaching and deterioration of catalytic activity. Furthermore, the filtration experiment together with the UV-Vis analysis of the filtrate solution sufficiently demonstrate that the catalytic species were well confined and no significant catalyst leaching could be detected during the reaction. In our opinion, the decreased activity might be relevant to the deterioration of the Hoveyda-Grubbs^{2nd} catalyst itself that arising from molecular decomposition or dimerization, which is preliminarily supported by the catalyst color change from green to gray [Ref. 67 Hong, S. H., Wenzel, A. G., Salguero, T. T., Day, M. W., Grubbs, R. H. Decomposition of Ruthenium Olefin Metathesis Catalysts. *J. Am. Chem. Soc.* 129, 7961-7968 (2007).]. In our revised manuscript, we have improved the analysis after

reaction as *“Moreover, our solid Grubbs/CALB@MOF catalyst could be easily recycled at least five times with remaining 79.3% catalytic activity (Fig. 6f). After reaction, the morphology and well-defined multi-compartmental microstructure are essentially unchanged, except for a slight degradation of the MOF structure, as confirmed by the SEM images and XRD patterns (Figs. 6g and h). The decreased activity might be relevant to the deterioration of Grubbs molecular catalyst caused by their decomposition or dimerization under reaction conditions⁶⁷.”*

In fact, to reveal the enhanced catalytic performance of our integrated Grubbs/CALB@MOF catalyst, we had performed comparisons of several representative systems, including the homogeneous counterparts (Grubbs/CALB), mechanical mixture of the separated catalysts (Grubbs@MOF/CALB@MOF) and traditional solvothermal immobilization system (Grubbs/CALB@traditional MOF). It was demonstrated that our cascade catalyst exhibited much better activity and selectivity than all of the other three systems. Here, to further confirm this, another synthetic system was also conducted according to your suggestion, where the Grubbs catalyst and CALB enzyme were together immobilized into the meso-channels of SBA-15 (a representative mesoporous silica support). As shown below (Fig. 6d in our revised manuscript), under identical conditions, a final conversion of 17.0% with 49.0% selectivity was obtained for Grubbs/CALB@SBA-15, and the corresponding CE value was calculated to be 5.3 mol mol⁻¹ h⁻¹. These results further confirmed the superior performance of our integrated Grubbs/CALB@MOF catalyst. In our revised manuscript, we have improved the above comparison results as *“To highlight as well as clarify this activity, several contrast experiments, including the homogeneous counterparts (Grubbs/CALB), mechanical mixture of the separated catalysts (Grubbs@MOF/CALB@MOF), traditional solvothermal immobilization system (Grubbs/CALB@traditional MOF) and mesoporous SBA-15 immobilized synthetic system (Grubbs/CALB@SBA-15) were also performed. Obviously, Grubbs/CALB@MOF exhibited much better activity and selectivity than all of the other four systems. As shown in Figs. 6d1 and 6d2, only 12.5, 25.6, 17.0% conversions and 16.3, 60.9, 49% selectivities were obtained for the free Grubbs/CALB, Grubbs@MOF/CALB@MOF and Grubbs/CALB@SBA-15, while almost no reaction occurred for the Grubbs/CALB@traditional MOF under identical reaction conditions. To better evaluate such distinct catalytic results, catalysis efficiency (CE) of these systems was also estimated based on the reaction profiles. As shown in Fig. 6d3, the integrated Grubbs/CALB@MOF affords a CE value of 21.5 mol mol⁻¹ h⁻¹, which is 5.81, 2.24 and 4.06-fold higher than that of free Grubbs/CALB system (3.7 mol mol⁻¹ h⁻¹), Grubbs@MOF/CALB@MOF system (9.6 mol mol⁻¹ h⁻¹) and Grubbs/CALB@SBA-15 (5.3 mol mol⁻¹ h⁻¹), respectively.”* and *“As for the immobilized synthetic system Grubbs/CALB@SBA-15, the relatively poor activity might be related with its low loading efficiency and hydrophilic channel properties, which is unfavorable for the catalytic activity of both Grubb’ catalyst and CALB species.”* Meanwhile, the detailed synthesis procedure of Grubbs/CALB@SBA-15 is also added in the Supplementary Information as *“For the synthesis of mesoporous SBA-15 immobilized system, 5 mg*

Grubbs catalyst (dissolved in 1.5 mL ethanol) together with 4 mL of CALB solution (0.1 mg mL⁻¹, PBS, pH = 8.0, 50 mM) were mixed with 0.2 g of mesoporous SBA-15. After slowly rotating at 35 °C overnight, the solid material was isolated and the obtained material was dried under vacuum. The obtained material is denoted as Grubbs/CALB@SBA-15.”

Figure D (Fig. 6d in the revised manuscript). Catalytic performance of the cascade reaction in different systems, including Grubbs/CALB@MOF, Grubbs/CALB, Grubbs@MOF/CALB@MOF, Grubbs/CALB@traditional MOF and Grubbs/CALB@SBA-15. Kinetic plots (**d₁**), selectivity of aimed CA product (**d₂**), and calculated catalysis efficiency (**d₃**).

Reviewer 2

The authors present a novel concept to synthesize MOF-walled microreactors for cascade reactions by employing a Pickering double emulsion “oil-on-water-on-oil” route. The basic idea is to prepare an emulsion@emulsion system in which MOF “walls” are grown inside an emulsion, which then leads to a microreactor system consisting of small, hollow MOF spheres inside a larger MOF sphere. By this route, both the smaller and the larger cavities can separately (i.e. consecutively) be “filled by different catalysts with the aim to perform cascade catalysis. The authors show that they are able to finetune and to control the size and density of the inner and outer microreactor by a multitude of different methods. This concept of synthesizing such kind of MOF@MOF materials is very innovative since it allows for a precise incorporation of each MOF cavity with different catalysts, which are spatially separated and thus the reactions that are catalyzed by the spatially separated catalysts should not interfere negatively or inhibit each other during the desired catalytic cascade reaction. In principle, I am quite fascinated by the beauty of the concept and in favor of the manuscript; nevertheless, there are some points regarding the significance of the catalytic data and the selection of the model reactions for demonstration of the system’s performance which should be clarified or substantiated before a decision on accepting the paper for publication can be made:

Response: We are deeply thankful for the supportive comments, careful review and valuable suggestions. The proposed issues and suggestions are really favorable for us to improve our manuscript. The questions are answered point-by-point as follows.

1. The kinetics of the formation of the MOF walls and the control over their thickness are somewhat unclear, could the authors be more specific about this.

Response: In fact, to explain the kinetics of the formation of the MOF walls, we had recorded the interfacial growth process of MOF layers and made description as “Moreover, since the fast coordination process of metal ions and organic ligands (Supplementary Fig. 14 in our revised manuscript), a solid MOF layer is formed quickly at the inner O/W and outer W/O interfaces (interaction iii), which further rigidifies the thermodynamically unstable droplets. Following the continuous interfacial growth of crystals, a robust MOF skeleton with interconnected tissue-like structure can be produced.” Although such an interfacial process occurred rapidly (as evidenced by the formation of reinforced droplets within only 5 min), a small part of the microreactors would be structurally destroyed in polar solvents in such a relatively short time, resulting in the lower production of materials. As the interfacial growth time is prolonged, a robust MOF skeleton can be produced. These results demonstrated that a suitable growth time was required to achieve the rigid and robust complex superstructure.

As for the control over their thickness, according to your suggestion, we have studied the effect of precursor concentration on the formed MOF layer. As shown below (Figure

9 in our revised Supplementary Information), taking a representative MOF-74 as an example, we investigated the interfacial growth by tuning the initial metal salt dosage from 1.37 to 1.03, 0.86 and 0.34 mmol, while the organic linker in inner droplets was always kept constant at 0.012 mmol and in excess in outer oil phase respectively. As expected, the multiple liquid-liquid interface could be solidified by a MOF layer at all tested dosages. Similar microcapsules with a particle size of $30 \pm 10 \mu\text{m}$ could be obtained after drying, of which the hierarchically multi-compartmental architectures are clearly visible. A magnified cross-sectional view of the inner and outer boundary shows that the interfacially formed MOF layer is dense and continuous. The thickness of the outer MOF layer gradually increased from 40 to 200, 370 and 430 nm, while increasing the dosage of metal salt from 0.34 to 0.86, 1.03 and 1.37 mmol. Meanwhile, the shell thickness of sub-compartments also increased from 30 to 100, 160 and 280 nm. This is understandable, since the dosage of coordination precursors directly determines the quantity of MOF skeletons formed at the interface, leading to distinct MOF layers with varied thicknesses degrees. In our revised manuscript, we have added this result in Supplementary Fig. 9, and made descriptions as “*Moreover, we found that the dosage of coordination precursors directly determined the formed MOF skeletons at the interface, leading to distinct inner and outer MOF layers with varied thicknesses (Supplementary Fig. 9).*”

Figure E (Supplementary Figure 9 in the revised version). SEM images of the multi-compartmental MOF-74 microreactors as a function of metal ion dosage, including 0.34, 0.86, 1.03 and 1.37 mmol. (a₁-a₄, b₁-b₄) Observations at different magnifications, (c₁-c₄) a single broken microparticle, (d₁-d₄) high-magnified observation of the interior structures and outer boundary.

2. The 1st catalytic cascade reaction, i.e. the ring closing metathesis (with a Grubbs catalyst in the small reactors)/transesterification (with a CALB enzyme in the larger reactors) is, as the authors pointed out, kinetically controlled: the 1st reaction is much faster than the 2nd one, which can be adjusted by increasing the concentration of the CALB catalyst. Nevertheless, this does not seem to be a unique feature of the authors' catalyst system; after all, the fully homogeneous Grubbs/CALB system does not perform significantly worse and it has not been optimized. Thus, it is not convincing to state that there is an "intrinsic incompatibility" of these two catalysts in the homogeneous phase (l. 413, p.20); this is not at all shown in the referenced Supp. Fig. 29 of the SI on p. S33. The initially low conversion/selectivity of the product could be interpreted in a way that vinyl acetate (VA) also undergoes metathesis with another VA molecule or with 1,6-heptadiene-4-ol, which could mean that the selectivity of the reaction is not controlled by the MOF@MOF microreactor system. Only an increase of the CALB conc. can lead to a faster decrease of the VA conc. and thereby suppress side reactions. In this regard, one could pose a provocative question: why does it require such a sophisticated [Catalyst 1@MOF]@[Catalyst 2@MOF] composite if both reactions can proceed simultaneously even if both catalysts can exist in one phase? What is the key unique feature of the system – beyond a somewhat faster and more efficient reaction?

Response: Firstly, we want to stress that the used CALB enzyme and Grubbs catalyst are not completely inactivated with each other in our study, which is quite different from other intrinsic incompatible systems like acid and base, oxidizing and reducing agents. The "intrinsic incompatibility" mentioned in our manuscript refers to the decreased catalytic activity after mixing two different world catalytic species together. Previously, to demonstrate the incompatibility of Grubbs catalyst and CALB, we had conducted a homogeneous single-step ring-closing metathesis reaction (Supp. Fig. 29 of the SI in our previous version), which performed smoothly but drastically decreased in the cascade reaction after the addition of CALB enzyme. As the reviewer mentioned, this decreased activity might be also interpreted in a way that vinyl acetate (VA) also undergoes metathesis with another VA molecule or with 1,6-heptadiene-4-ol, which will make readers puzzled. Herein, to clarify this, we further conducted the following experiments:

(1) For better comparison, we examined the catalytic performance of Grubbs catalyst for single-step ring-closing metathesis reaction in the presence of CALB enzyme. As shown below, the final conversion decreased from 100% to 54.2% after the addition of CALB enzyme into Grubbs catalyst solution, which indicates that CALB enzyme truly caused a decreased catalytic activity of Grubbs catalyst. We have added this result in our revised Supporting Information as Supplementary Fig. 30.

Figure F (Supplementary Figure 30 in the revised version). Kinetic plots for ring-closing metathesis of 1,6-heptadien-4-ol over the homogenous mixture of Grubbs/CALB and pure Grubbs' catalyst. Reaction conditions: 0.1 mmol 1,6-heptadien-4-ol, 1 mL *n*-hexane, 5 mg Grubbs' catalyst or the mixture of 5 mg Grubbs' catalyst and 0.24 mg enzyme, 45 °C.

(2) To further verify the intrinsic incompatibility and optimize the homogeneous Grubbs/CALB system, we also studied the catalytic performance of Grubbs/CALB under different mass ratios according to your suggestion. As shown below, when the amount of CALB was increased from 0.4 to 0.8 and 1.6 mg, while keeping the mass of Grubbs' catalyst at 5 mg, the overall conversion gradually increased from 12.5% to 21.6% and 30.0%, while the selectivity of CA improved from 16.3% to 20.3% and 28.0%. Meanwhile, the corresponding CE was calculated to be 3.7, 7.3 and 10.2 mol mol⁻¹ h⁻¹. Although the improved activity and selectivity were exhibited as increasing the amount of CALB, the optimized catalytic efficiency of homogeneous Grubbs/CALB system is still lower than that of homogenous Grubbs catalyst, indicating that incompatibility between different catalytic species leads to the poor activity of homogeneous system. Given the fact that this result is similar with Fig. 6d₁ and the paper is already too long, we didn't add these results in our revised manuscript after careful consideration.

Figure G. Catalytic results of homogenous Grubbs/CALB system with different mass ratios of enzyme and molecular catalyst. a, Kinetic plots. b, Selectivity of aimed CA

product. c, Calculated catalysis efficiency.

(3) Furthermore, we studied the cross-metathesis of vinyl acetate (VA) with another VA molecule in the presence of Grubbs catalyst, but almost no reaction occurred under identical reaction conditions. As for the metathesis of VA with 1,6-heptadiene-4-ol, in fact, we had performed the related experiments and made descriptions as “*As shown in Supplementary Fig. 28 (Supplementary Fig. 29 in our revised manuscript), only RCM or transesterification reaction occurred in the presence of Grubbs@MOF or CALB@MOF, and the RCM step proceeded much faster than transesterification, indicating that the cascade reaction mainly happened by route I.*” Also, the ring-closing metathesis (RCM) occurs much faster than cross-metathesis (CM) for dienes due to lowest energy principle. In our opinion, since the enzymatic step is much slower than the initial step motivated by Grubb’ catalyst, the conversion of the cascade reaction is mainly dominated by the CALB enzyme, thus leading to gradual increase of conversions with varying enzyme contents.

Therefore, considering the intrinsic incompatibility of different world catalytic species, we proposed a Pickering double emulsion-directed interfacial synthesis method for efficiently constructing multi-compartmental crystalline MOF microreactors, which can be utilized as a powerful platform for spatially controlled organization of incompatible functionalities inside the tailored synthetic compartments for operating concurrent chemo-enzymatic reaction. Furthermore, in the view of green and sustainable chemistry, this perspective is focused mainly on rational preparation of solid multi-compartmental MOF microreactors under mild conditions for heterogeneous catalysis. It was demonstrated that superior catalytic properties together with the excellent recyclability and structural stability were exhibited for our MOF microreactors in the cascade reaction.

3. Does an alteration of the substrates (e.g. by polarity) have an influence on the reaction kinetics (e.g. by different diffusion rates)? Do the surfactants of the SiO₂ NPs play a role in the interaction with the substrates?

Response: To reveal the influence of an alteration of substrates on the reaction kinetics, another two substrates for the one-pot cascade reaction based on the Grubb’ catalyst and CALB lipase have been studied. As shown below, under identical reaction conditions, for substrate 4-methyl-1,6-heptadien-4-ol (Reaction I), a conversion of 80.1% together with a selectivity of 56.3% were obtained within 150 min over Grubbs/CALB@MOF (200 mg, the loading content of CALB and Grubbs’ catalyst were 8 mg g⁻¹ and 25 mg g⁻¹ respectively) in the one-pot ring-closing metathesis (RCM)/transesterification cascade reaction. Compared with previously studied 1, 6-heptadien-4-ol (as shown in Figure 6e₁ and e₂, 56.0% conversion and 94.6% selectivity within 90 min, Grubbs/CALB@MOF loading with 8 mg g⁻¹ CALB and 25 mg g⁻¹ Grubbs’ catalyst), the conversion is higher, but the selectivity is much lower. This result can be ascribed to the excellent catalytic activity of Grubbs’ catalyst towards substrate 4-methyl-1,6-heptadien-4-ol and significantly lower catalytic activity of CALB enzyme towards the intermediate product 1-methyl-3-cyclopenten-1-ol. We had attempted to

improve the loading content of CALB enzyme so as to boost the final selectivity, but too much loading will disturb the emulsification process and thus fail to obtain the final solid catalyst. Also, we extended the integrated Grubbs/CALB@MOF to the ring-closing metathesis (RCM)/hydrolysis cascade reaction taking diethyl diallylmalonate as a substrate (Reaction II), and a conversion of 98.0% with a selectivity of 29.0% (200 mg, the loading content of CALB and Grubbs' catalyst were 8 mg g⁻¹ and 25 mg g⁻¹ respectively) was achieved. The inferior selectivity might be ascribed to its biphasic catalysis system, which lead to much lower catalytic activity of the enzymatic hydrolysis step. Given the above, the extended catalytic results demonstrated the generic applicability of our Grubbs/CALB@MOF in the chemo-enzymatic cascade reactions, but the catalytic activity and selectivity are quite different from each other by altering the substrates. We have added this result in Supplementary Fig. 32, and made related descriptions in our revised manuscript as “*Furthermore, our integrated catalyst Grubbs/CALB@MOF could be expanded to the chemo-enzymatic cascade reaction of other substrates. As shown in Supplementary Fig. 32, for substrate 4-methyl-1,6-heptadien-4-ol (Reaction I), a conversion of 80.1% together with a selectivity of 56.3% were obtained in the one-pot RCM/transesterification cascade reaction. Meanwhile, the solid Grubbs/CALB@MOF was applied to the RCM/hydrolysis cascade reaction with diethyl diallylmalonate as a substrate (Reaction II), achieving a conversion of 98.0% with a selectivity of 29.0%. This inferior selectivity might be ascribed to the biphasic catalysis system, which lead to much lower catalytic activity of the enzymatic hydrolysis step. These extended catalytic results further demonstrated the generic applicability of our Grubbs/CALB@MOF in chemo-enzymatic cascade reactions.*”

Figure H. (Supplementary Figure 32 in the revised version). Catalytic performance of Grubbs/CALB@MOF in the one-pot cascade reaction of different substrates. a, Reaction networks. b, Kinetic plots. c, Selectivity of aimed product.

Reaction conditions: For reaction I, 0.1 mmol 4-methyl-1,6-heptadien-4-ol, 1 mL dichloromethane, 200 mg Grubbs/CALB@MOF (the loading content of CALB and Grubbs' catalyst were 8 mg g⁻¹ and 25 mg g⁻¹ respectively), 60 °C. For reaction II, 0.1 mmol diethyl diallylmalonate, 1 mL n-hexane, 300 µL deionized water, 200 mg Grubbs/CALB@MOF (the loading content of CALB and Grubbs' catalyst were 8 mg g⁻¹ and 25 mg g⁻¹ respectively), 45 °C.

As for the role of SiO₂ nanoparticles, they are only utilized as interfacially active species to form stabilized emulsion droplets, which adsorb at the liquid-liquid interface and prevent droplet coalescence. These SiO₂ nanoparticles are inert and can't drive the cascade reaction. Moreover, no specific interactions are existed between the substrates and SiO₂ nanoparticles. Therefore, in our opinion, the surfactants of the SiO₂ NPs didn't play a role in the interaction with the substrates.

4. Fig. 6f/recycling: the Fig. shows the conversion steadily decreasing with each run, which, together with the XRDs in Fig. 6h, points to a degradation of the MOF structure (and possible catalyst leaching due to a partial collapse of the MOF walls), which in turn speaks against a good stability (as stated on p.22, l.458/459). Can the authors comment on this discrepancy? Has the biphasic system been stirred during the reaction? Is there a difference in MOF stability after the catalytic reaction for different MOF types?

Response: As the reviewer mentioned, the conversion steadily decreased with each run during the recycling tests, and a degradation of the MOF structure might be happened from the XRD patterns in Fig. 6h. However, such a structure degradation is relatively slight, which is not enough to lead catalyst leaching and deterioration of catalytic activity. Firstly, from the XRD patterns, we can see that most of the peaks are virtually unchanged, and the SEM images also confirm the good retain of morphology and multi-compartmental microstructure after recycling. Secondly, the filtration experiment together with the UV-Vis analysis of the filtrate solution sufficiently demonstrate that the catalytic species were well confined and no significant catalyst leaching could be detected during the reaction. In our opinion, the decreased activity might be relevant to the deterioration of the Hoveyda-Grubbs^{2nd} catalyst itself that arising from molecular decomposition or dimerization, which has been demonstrated by some literature reports [Ref. 67 Hong, S. H., Wenzel, A. G., Salguero, T. T., Day, M. W., Grubbs, R. H. Decomposition of Ruthenium Olefin Metathesis Catalysts. *J. Am. Chem. Soc.* 129, 7961-7968 (2007).]. In our revised manuscript, we have improved the analysis after reaction as *“Moreover, our solid Grubbs/CALB@MOF catalyst could be easily recycled at least five times with remaining 79.3% catalytic activity (Fig. 6f). After reaction, the morphology and well-defined multi-compartmental microstructure are essentially unchanged, except for a slight degradation of the MOF structure, as confirmed by the SEM images and XRD patterns (Figs. 6g and h). The decreased activity might be relevant to the deterioration of Grubbs molecular catalyst caused by their decomposition or dimerization under reaction conditions⁶⁷.”*

For evaluating the catalytic activity, all of the reactions were conducted under a

stirring rate of 300 rpm. We have added this description as “*After fast sealing under N₂ atmosphere, the reaction tube was kept at 45 °C under a stirring rate of 300 rpm.*” in our revised Supplementary Information.

Finally, to reveal the possible difference in MOF stability after the catalytic reaction, we have studied three different types of MOF microreactors, including Ni-MOF-74-II, MIL-100 and HKUST-1. The catalytic performance of these MOF microreactors in the Grubb’ catalyst/CALB lipase driven olefin metathesis/transesterification cascade reaction were also evaluated, and the corresponding results are shown as below. Clearly, the catalytic performances of these different typed MOF-microreactors are quite different from each other. For example, under identical conditions, only 45.9, 49.0% conversions and 0.4, 31.2% selectivities were obtained for MIL-88A and HKUST-1 microreactors. Meanwhile, the corresponding CE was calculated to be 18.55 and 21.15 mol mol⁻¹ h⁻¹. The serious deterioration of product selectivity in MIL-88A microreactor could be ascribed to the impacts of added Fe³⁺ salt on enzymes during their synthetic processes, which has been demonstrated by our previous report [Ref. 50 Tian, D. et al. Pickering-Droplet-Derived MOF Microreactors for Continuous-Flow Biocatalysis with Size Selectivity. *J. Am. Chem. Soc.* 143, 16641-16652 (2021).]. As for Ni-MOF-74-II microreactor, 21.5% of final conversion and 99% of CA selectivity were obtained. This relatively poor catalysis efficiency might be related with its larger pore size of MOF layers (theoretical pore size is 2.2 nm), which is unfavorable to achieve effective encapsulation and confinement of metal complex catalyst inside this MOF structure. After reaction, the microspherical morphology and multi-compartmental interior structure were all well reserved, except that a small portion of the crystal MOF layers are destroyed as confirmed by the slightly changed XRD patterns. These stability observations are similar with our described Grubbs/CALB@MOF (Ni-MOF-74-I) catalyst. Moreover, this paper is already too long. Therefore, after careful consideration, we didn’t add the related results in our revised manuscript.

Figure I. Catalytic results of different types of Grubbs/CALB@MOF microreactors, including Ni-MOF-74-II, MIL-88A and HKUST-1. a, Kinetic plots. b, Selectivity of aimed CA product. c, Calculated catalysis efficiency. (d-f) SEM images and XRD patterns of the multi-compartmental MOF microreactors after reaction. (d₁-f₁) Observations at low magnification, (d₂-f₂) a single broken microparticle, (d₃-f₃) high-magnified observation of the interior structures, (d₄-f₄) XRD pattern.

Reviewer 3

This study presented a nice method of constructing multi-compartmental MOF microreactors from Pickering double emulsions for chemo-enzymatic cascade catalysis. The structure of the multi-compartmental MOF microreactors is novel and important for the high efficiency of chemo-enzymatic cascade catalysis. The Grubb' catalyst/CALB and GOx/Fe systems encapsulated in the microreactors exhibited 2.24-5.81 fold enhanced catalytic efficiency. And the authors found that the enhanced efficiency is a result of spatial confinement of two incompatible catalysts and the increased transportation efficiency of the reaction intermediate. Overall, this is a well-organized manuscript which can be accepted after addressing minor points.

Response: Thank you very much for your positive comments. We have revised our manuscript according to your suggestions as follows.

1. When discussing the substrate channeling effect of the reaction intermediate between CALB and the Grubb' catalyst, the following reference would be helpful: Nano Letters 2022, 22, 5029-5036.

Response: The suggested paper is about constructing multienzyme system in amorphous metal-organic frameworks, and a nice diffusion-reaction model was proposed to elucidate the relationship between enzyme distance and activity. The work is meaningful and really helpful for us to understand the substrate channeling effects of the reaction intermediate between CALB and the Grubb' catalyst, thus we cited this paper as "Ref.66 Zhang, Y. Y., Xu, L. J., Ge, J. Multienzyme System in Amorphous Metal-Organic Frameworks for Intracellular Lactate Detection. Nano Lett. 22, 5029-5036 (2022)." in our revised manuscript.

2. In the cascade reaction, the Grubb' catalyst and CALB may work at different optimal temperatures. Did the authors try to investigate the effect of reaction temperature of this cascade reaction?

Response: We agree with you that the Grubb' catalyst and CALB may work at different optimal temperatures in the cascade reaction. So, we studied the impacts of reaction temperature on the catalysis efficiency in our revised manuscript.

Firstly, we studied the influence of reaction temperature on the catalytic performances of Grubbs@MOF and CALB@MOF for the single-step reactions of chemo-enzymatic cascade reaction. As shown below, both of these two solid catalysts exhibited increased activity along with the increasing of reaction temperature. In the ring-closing metathesis of 1, 6-heptadien-4-ol, 31.3, 71.0 and 86.0% of conversions were got for Grubbs@MOF at temperatures of 25, 45 and 70 °C. When CALB@MOF was used for the transesterification of 3-cyclopenten-1-ol, 25.0, 75.2, and 81.6% conversions were achieved at reaction temperatures of 25, 45 and 70 °C respectively.

Figure J. Kinetic plots for the single-step reactions over Grubbs@MOF or CALB@MOF. a, The ring-closing metathesis of 1, 6-heptadien-4-ol over Grubbs@MOF. b, The transesterification of 3-cyclopenten-1-ol over CALB@MOF.

Given the above results, we then examined the catalytic performance of Grubbs/CALB@MOF for the one-pot cascade reaction at different temperatures. As shown below, along with the increasing of reaction temperature from 25 to 45 and 70 °C, the overall conversion gradually increased from 21.0% to 48.9% and 84.2%, while the selectivity of CA drastically improved from 19.8% to 74.6% and 86.3%. Meanwhile, the corresponding CE was calculated to be 6.2, 21.5 and 32.7 mol mol⁻¹ h⁻¹ respectively. However, although better activity was achieved at a higher temperature of 70 °C, we still decided to carry out the reactions at 45 °C, since the good stability of enzymes under mild working conditions. In our revised manuscript, we have added the corresponding results as Supplementary Fig. 31 and made related descriptions as “*Moreover, we also studied the impacts of reaction temperature on the catalysis efficiency to optimize the reaction conditions. As shown in Supplementary Fig. 31, the reaction temperature shows a significant influence on the catalytic performance of chemo-enzymatic cascade reaction. Along with the increasing of reaction temperature from 25 to 45 and 70 °C, the overall conversion gradually increased from 21.0 to 48.9 and 84.2%, while the selectivity of CA drastically improved from 19.8 to 74.6 and 86.3%. Meanwhile, the corresponding CE was calculated to be 6.2, 21.5 and 32.7 mol mol⁻¹ h⁻¹. Although better activity was achieved at a higher temperature of 70 °C, we decided to carry out the reactions at 45 °C, since the good stability of enzymes under mild working conditions.*”

Figure K (Supplementary Figure 31 in the revised version). Catalytic results of Grubbs/CALB@MOF microreactors for the ring-closing metathesis/transesterification cascade reaction at different temperatures. a, Reaction networks. b, Kinetic plots. c, Selectivity of aimed CA product. d, Calculated catalysis efficiency.

3. Page 22, “Since the enzymatic step is much slower than the initial step motivated by Grubb’ catalyst, the conversion of the cascade reaction is mainly dominated by the Grubb’ catalyst, thus leading to gradual increase of conversions with varying enzyme contents.” should be “...is mainly dominated by enzyme...”.

Response: We have corrected this clerical error in our revised manuscript as “*Since the enzymatic step is much slower than the initial step motivated by Grubb’ catalyst, the conversion of the cascade reaction is mainly dominated by the enzyme, thus leading to gradual increase of conversions with varying enzyme contents.*” Thank you so much for your careful review.

Reviewers' Comments:

Reviewer #1:

Remarks to the Author:

I read the revised manuscript, and found that the author has made accurate and reasonable responses together with effective improvements to my previous comments. I feel the manuscript has significantly and satisfactorily improved and still stand by my initial judgement that the work is highly innovative and of high quality. In my opinion, this manuscript can be accepted directly for publication in Nature Communication as it is.

Reviewer #2:

Remarks to the Author:

The authors provided convincing revisions and discussions of basically all the raised issues. I am happy to recommend acceptance of the revised manuscript as is. Congratulations to their interesting results.

Reviewer #3:

Remarks to the Author:

The authors have addressed my concerns and the manuscript can be accepted for publication.

Thanks for your suggestions and supportive comments from all reviewers.